# Adversarial Attack Generation Empowered by Min-Max Optimization

**Jingkang Wang**[1,2*]     **Tianyun Zhang**[3*]     **Sijia Liu**[4,5]
**Pin-Yu Chen**[5]     **Jiacen Xu**[6]     **Makan Fardad**[7]     **Bo Li**[8]

University of Toronto[1], Vector Institute[2], Cleveland State University[3]
Michigan State University[4], MIT-IBM Watson AI Lab, IBM Research[5]
University of California, Irvine[6], Syracuse University[7]
University of Illinois at Urbana-Champaign[8]

## Abstract

The worst-case training principle that minimizes the maximal adversarial loss, also known as adversarial training (AT), has shown to be a state-of-the-art approach for enhancing adversarial robustness. Nevertheless, min-max optimization beyond the purpose of AT has not been rigorously explored in the adversarial context. In this paper, we show how a general framework of min-max optimization over multiple domains can be leveraged to advance the design of different types of adversarial attacks. In particular, given a set of risk sources, minimizing the worst-case attack loss can be reformulated as a min-max problem by introducing domain weights that are maximized over the probability simplex of the domain set. We showcase this unified framework in three attack generation problems – attacking model ensembles, devising universal perturbation under multiple inputs, and crafting attacks resilient to data transformations. Extensive experiments demonstrate that our approach leads to substantial attack improvement over the existing heuristic strategies as well as robustness improvement over state-of-the-art defense methods trained to be robust against multiple perturbation types. Furthermore, we find that the self-adjusted domain weights learned from our min-max framework can provide a holistic tool to explain the difficulty level of attack across domains. Code is available at https://github.com/wangjksjtu/minmax-adv.

## 1   Introduction

Training a machine learning model that is capable of assuring its worst-case performance against possible adversaries given a specified threat model is a fundamental and challenging problem, especially for deep neural networks (DNNs) [64, 22, 13, 69, 70]. A common practice to train an adversarially robust model is based on a specific form of min-max training, known as *adversarial training* (AT) [22, 40], where the minimization step learns model weights under the adversarial loss constructed at the maximization step in an alternative training fashion. In practice, AT has achieved the state-of-the-art defense performance against $\ell_p$-norm-ball input perturbations [3].

Although the min-max principle is widely used in AT and its variants [40, 59, 76, 65], few work has studied its power in attack generation. Thus, we ask: *Beyond AT, can other types of min-max formulation and optimization techniques advance the research in adversarial attack generation?* In this paper, we give an affirmative answer corroborated by the substantial performance gain and the ability of self-learned risk interpretation using our proposed min-max framework on several tasks for adversarial attack.

---

[*]Equal contributions.

35th Conference on Neural Information Processing Systems (NeurIPS 2021).

We demonstrate the utility of a general formulation for minimizing the maximal loss induced from a set of risk sources (domains). Our considered min-max formulation is fundamentally different from AT, as our maximization step is taken over the probability simplex of the set of domains. Moreover, we show that many problem setups in adversarial attacks can in fact be reformulated under this general min-max framework, including attacking model ensembles [66, 34], devising universal perturbation to input samples [44] and data transformations [6, 10]. However, current methods for solving these tasks often rely on simple heuristics (e.g., uniform averaging), resulting in significant performance drops when comparing to our proposed min-max optimization framework.

**Contributions**   ① With the aid of min-max optimization, we propose a unified alternating one-step projected gradient descent-ascent (APGDA) attack method, which can readily be specified to generate model ensemble attack, universal attack over multiple images, and robust attack over data transformations. ② In theory, we show that APGDA has an $O(1/T)$ convergence rate, where $T$ is the number of iterations. In practice, we show that APGDA obtains 17.48%, 35.21% and 9.39% improvement on average compared with conventional min-only PGD attack methods on CIFAR-10. ③ More importantly, we demonstrate that by tracking the learnable weighting factors associated with multiple domains, our method can provide tools for self-adjusted importance assessment on the mixed learning tasks. ④ Finally, we adapt the idea of the domain weights into a defense setting [65], where multiple $\ell_p$-norm perturbations are generated, and achieve superior performance as well as intepretability.

## 1.1   Related work

Recent studies have identified that DNNs are highly vulnerable to adversarial manipulations in various applications [64, 12, 27, 33, 26, 14, 77, 20, 15, 31], thus leading to an arms race between adversarial attacks [13, 3, 23, 48, 45, 72, 1, 18] and defenses [40, 59, 76, 65, 42, 71, 74, 68, 53, 16]. One intriguing property of adversarial examples is the transferability across multiple domains [36, 67, 47, 62], which indicates a more challenging yet promising research direction – devising universal adversarial perturbations over model ensembles [66, 34], input samples [44, 43, 56] and data transformations [3, 6, 10].

Besides, many recent works started to produce physical realizable perturbations that expose real world threats. The most popular approach [4, 21], as known as Expectation Over Transformation (EOT), is to train the attack under different data transformation (e.g., different view angles and distances). However, current approaches suffer from a significant performance loss for resting on the uniform averaging strategy or heuristic weighting schemes [34, 56]. We will compare these works with our min-max method in Sec. 4. As a natural extension following min-max attack, we study the generalized AT under multiple perturbations [65, 2, 28, 17]. Finally, our min-max framework is adapted and inspired by previous literature on robust optimization over multiple domains [50, 51, 38, 37].

To our best knowledge, only few works leverage min-max principle for adversarial attack generation while the idea of producing the worst-case example across multiple domains is quite natural. Specifically, [7] considered the non-interactive blackbox adversary setting and proposed a framework that models the crafting of adversarial examples as a min-max game between a generator of attacks and a classifier. [57] introduced a min-max based adaptive attacker's objective to craft perturbation so that it simultaneously evades detection and causes misclassification. Inspired by our work, the min-max formulation has also been extended to zero-order blackbox attacks [35] and physically realizable attacks [73, Adversarial T-shirt]. We hope our unified formulation can stimulate further research on applying min-max principle and interpretable domain weights in more attack generation tasks that involve in evading multiple systems.

## 2   Min-Max Across Domains

Consider $K$ loss functions $\{F_i(\mathbf{v})\}$ (each of which is defined on a learning domain), the problem of robust learning over $K$ domains can be formulated as [50, 51, 38]

$$\underset{\mathbf{v}\in\mathcal{V}}{\text{minimize}}\ \underset{\mathbf{w}\in\mathcal{P}}{\text{maximize}}\quad \sum_{i=1}^{K} w_i F_i(\mathbf{v}), \tag{1}$$

where $\mathbf{v}$ and $\mathbf{w}$ are optimization variables, $\mathcal{V}$ is a constraint set, and $\mathcal{P}$ denotes the probability simplex $\mathcal{P} = \{\mathbf{w} \mid \mathbf{1}^T\mathbf{w} = 1, w_i \in [0, 1], \forall i\}$. Since the inner maximization problem in (1) is a linear function

of $\mathbf{w}$ over the probabilistic simplex, problem (1) is thus equivalent to

$$\underset{\mathbf{v} \in \mathcal{V}}{\text{minimize}} \ \underset{i \in [K]}{\text{maximize}} \quad F_i(\mathbf{v}), \tag{2}$$

where $[K]$ denotes the integer set $\{1, 2, \ldots, K\}$.

**Benefit and Challenge from (1).** Compared to multi-task learning in a finite-sum formulation which minimizes $K$ losses on *average*, problem (1) provides consistently robust *worst-case* performance across all domains. This can be explained from the epigraph form of (2),

$$\underset{\mathbf{v} \in \mathcal{V}, t}{\text{minimize}} \ t, \quad \text{subject to } F_i(\mathbf{v}) \leq t, i \in [K], \tag{3}$$

where $t$ is an epigraph variable [8] that provides the $t$-level robustness at each domain.

In computation, the inner maximization problem of (1) always returns the one-hot value of $\mathbf{w}$, namely, $\mathbf{w} = \mathbf{e}_i$, where $\mathbf{e}_i$ is the $i$th standard basis vector, and $i = \arg\max_i\{F_i(\mathbf{v})\}$. However, this one-hot coding reduces the generalizability to other domains and induces instability of the learning procedure in practice. Such an issue is often mitigated by introducing a *strongly concave regularizer* in the inner maximization step to strike a balance between the average and the worst-case performance [38, 50].

**Regularized Formulation.** Following [50], we penalize the distance between the *worst-case* loss and the *average* loss over $K$ domains. This yields

$$\underset{\mathbf{v} \in \mathcal{V}}{\text{minimize}} \ \underset{\mathbf{w} \in \mathcal{P}}{\text{maximize}} \quad \sum_{i=1}^{K} w_i F_i(\mathbf{v}) - \tfrac{\gamma}{2}\|\mathbf{w} - \mathbf{1}/K\|_2^2, \tag{4}$$

where $\gamma > 0$ is a regularization parameter. As $\gamma \to 0$, problem (4) is equivalent to (1). By contrast, it becomes the finite-sum problem when $\gamma \to \infty$ since $\mathbf{w} \to \mathbf{1}/K$. *In this sense, the trainable $\mathbf{w}$ provides an essential indicator on the importance level of each domain.* The larger the weight is, the more important the domain is. We call $\mathbf{w}$ *domain weights* in this paper.

# 3 Min-Max Power in Attack Design

To the best of our knowledge, few work has studied the power of min-max in attack generation. In this section, we demonstrate how the unified min-max framework (4) fits into various attack settings. With the help of domain weights, our solution yields better empirical performance and explainability. Finally, we present the min-max algorithm with convergence analysis to craft robust perturbations against multiple domains.

## 3.1 A Unified Framework for Robust Adversarial Attacks

The general goal of adversarial attack is to craft an adversarial example $\mathbf{x}' = \mathbf{x}_0 + \boldsymbol{\delta} \in \mathbb{R}^d$ to mislead the prediction of machine learning (ML) or deep learning (DL) systems, where $\mathbf{x}_0$ denotes the natural example with the true label $t_0$, and $\boldsymbol{\delta}$ is known as *adversarial perturbation*, commonly subject to $\ell_p$-norm ($p \in \{0, 1, 2, \infty\}$) constraint $\mathcal{X} := \{\boldsymbol{\delta} \,|\, \|\boldsymbol{\delta}\|_p \leq \epsilon, \ \mathbf{x}_0 + \boldsymbol{\delta} \in [0, 1]^d\}$ for a given small number $\epsilon$. Here the $\ell_p$ norm enforces the similarity between $\mathbf{x}'$ and $\mathbf{x}_0$, and the input space of ML/DL systems is normalized to $[0, 1]^d$.

**Ensemble Attack over Multiple Models.** Consider $K$ ML/DL models $\{\mathcal{M}_i\}_{i=1}^{K}$, the goal is to find robust adversarial examples that can fool all $K$ models *simultaneously*. In this case, the notion of 'domain' in (4) is specified as 'model', and the objective function $F_i$ in (4) signifies the attack loss $f(\boldsymbol{\delta}; \mathbf{x}_0, y_0, \mathcal{M}_i)$ given the natural input $(\mathbf{x}_0, y_0)$ and the model $\mathcal{M}_i$. Thus, problem (4) becomes

$$\underset{\boldsymbol{\delta} \in \mathcal{X}}{\text{minimize}} \ \underset{\mathbf{w} \in \mathcal{P}}{\text{maximize}} \quad \sum_{i=1}^{K} w_i f(\boldsymbol{\delta}; \mathbf{x}_0, y_0, \mathcal{M}_i) - \tfrac{\gamma}{2}\|\mathbf{w} - \mathbf{1}/K\|_2^2, \tag{5}$$

where $\mathbf{w}$ encodes the difficulty level of attacking each model.

**Universal Perturbation over Multiple Examples.** Consider $K$ natural examples $\{(\mathbf{x}_i, y_i)\}_{i=1}^{K}$ and a single model $\mathcal{M}$, our goal is to find the universal perturbation $\boldsymbol{\delta}$ so that all the corrupted $K$ examples can fool $\mathcal{M}$. In this case, the notion of 'domain' in (4) is specified as 'example', and problem (4) becomes

$$\underset{\boldsymbol{\delta} \in \mathcal{X}}{\text{minimize}} \ \underset{\mathbf{w} \in \mathcal{P}}{\text{maximize}} \quad \sum_{i=1}^{K} w_i f(\boldsymbol{\delta}; \mathbf{x}_i, y_i, \mathcal{M}) - \tfrac{\gamma}{2}\|\mathbf{w} - \mathbf{1}/K\|_2^2, \tag{6}$$

where different from (5), $\mathbf{w}$ encodes the difficulty level of attacking each example.

**Adversarial Attack over Data Transformations.** Consider $K$ categories of data transformation $\{p_i\}$, e.g., rotation, lightening, and translation, our goal is to find the adversarial attack that is robust to data transformations. Such an attack setting is commonly applied to generate physical adversarial examples [5, 20]. Here the notion of 'domain' in (4) is specified as 'data transformer', and problem (4) becomes

$$\underset{\boldsymbol{\delta}\in\mathcal{X}}{\text{minimize}}\ \underset{\mathbf{w}\in\mathcal{P}}{\text{maximize}}\quad \sum_{i=1}^{K} w_i \mathbb{E}_{t\sim p_i}[f(t(\mathbf{x}_0+\boldsymbol{\delta}); y_0, \mathcal{M})] - \tfrac{\gamma}{2}\|\mathbf{w}-\mathbf{1}/K\|_2^2, \qquad (7)$$

where $\mathbb{E}_{t\sim p_i}[f(t(\mathbf{x}_0+\boldsymbol{\delta}); y_0, \mathcal{M})]$ denotes the attack loss under the distribution of data transformation $p_i$, and $\mathbf{w}$ encodes the difficulty level of attacking each type of transformed example $\mathbf{x}_0$. We remark that if $\mathbf{w} = \mathbf{1}/K$, then problem (7) reduces to the existing expectation of transformation (EOT) setup used for physical attack generation [5].

**Benefits of Min-Max Attack Generation with Learnable Domain Weights w:** We can interpret (5)-(7) as finding the *robust* adversarial attack against the *worst-case environment* that an adversary encounters, e.g., multiple victim models, data samples, and input transformations. The proposed min-max design of adversarial attacks leads to two main benefits. First, compared to the heuristic weighting strategy (e.g., clipping thresholds on the importance of individual attack losses [56]), our proposal is free of supervised manual adjustment on domain weights. Even by carefully tuning the heuristic weighting strategy, we find that our approach with self-adjusted $\mathbf{w}$ consistently outperforms the clipping strategy in [56] (see Table 2). Second, the learned domain weights can be used to assess the model robustness when facing different types of adversary. We refer readers to Figure 1c and Figure 6 for more details.

## 3.2 Min-Max Algorithm for Adversarial Attack Generation

We propose the **a**lternating **p**rojected **g**radient **d**escent-**a**scent (APGDA) method (Algorithm 1) to solve problem (4). For ease of presentation, we write problems (5), (6), (7) into the general form

$$\underset{\boldsymbol{\delta}\in\mathcal{X}}{\text{minimize}}\ \underset{\mathbf{w}\in\mathcal{P}}{\text{maximize}}\quad \sum_{i=1}^{K} w_i F_i(\boldsymbol{\delta}) - \tfrac{\gamma}{2}\|\mathbf{w}-\mathbf{1}/K\|_2^2,$$

where $F_i$ denotes the $i$th individual attack loss. We show that at each iteration, APGDA takes only one-step PGD for outer minimization and one-step projected gradient ascent for inner maximization.

---

**Algorithm 1** APGDA to solve problem (4)

1: Input: given $\mathbf{w}^{(0)}$ and $\boldsymbol{\delta}^{(0)}$.
2: **for** $t = 1, 2, \ldots, T$ **do**
3:   *outer min.*: fixing $\mathbf{w} = \mathbf{w}^{(t-1)}$, call PGD (8) to update $\boldsymbol{\delta}^{(t)}$
4:   *inner max.*: fixing $\boldsymbol{\delta} = \boldsymbol{\delta}^{(t)}$, update $\mathbf{w}^{(t)}$ with projected gradient ascent (9)
5: **end for**

---

**Outer Minimization** Considering $\mathbf{w} = \mathbf{w}^{(t-1)}$ and $F(\boldsymbol{\delta}) := \sum_{i=1}^{K} w_i^{(t-1)} F_i(\boldsymbol{\delta})$ in (4), we perform one-step PGD to update $\boldsymbol{\delta}$ at iteration $t$,

$$\boldsymbol{\delta}^{(t)} = \text{proj}_{\mathcal{X}}\left(\boldsymbol{\delta}^{(t-1)} - \alpha \nabla_{\boldsymbol{\delta}} F(\boldsymbol{\delta}^{(t-1)})\right), \qquad (8)$$

where $\text{proj}(\cdot)$ denotes the Euclidean projection operator, i.e., $\text{proj}_{\mathcal{X}}(\mathbf{a}) = \arg\min_{\mathbf{x}\in\mathcal{X}} \|\mathbf{x}-\mathbf{a}\|_2^2$ at the point $\mathbf{a}$, $\alpha > 0$ is a given learning rate, and $\nabla_{\boldsymbol{\delta}}$ denotes the first-order gradient w.r.t. $\boldsymbol{\delta}$. If $p = \infty$, then the projection function becomes the clip function. In Proposition 1, we derive the solution of $\text{proj}_{\mathcal{X}}(\mathbf{a})$ under different $\ell_p$ norms for $p \in \{0, 1, 2\}$.

**Proposition 1.** *Given a point* $\mathbf{a} \in \mathbb{R}^d$ *and a constraint set* $\mathcal{X} = \{\boldsymbol{\delta} | \|\boldsymbol{\delta}\|_p \leq \epsilon, \check{\mathbf{c}} \leq \boldsymbol{\delta} \leq \hat{\mathbf{c}}\}$, *the Euclidean projection* $\boldsymbol{\delta}^* = \text{proj}_{\mathcal{X}}(\mathbf{a})$ *has a closed-form solution when* $p \in \{0, 1, 2\}$, *where the specific form is given by Appendix A.*

**Inner Maximization** By fixing $\boldsymbol{\delta} = \boldsymbol{\delta}^{(t)}$ and letting $\psi(\mathbf{w}) := \sum_{i=1}^{K} w_i F_i(\boldsymbol{\delta}^{(t)}) - \tfrac{\gamma}{2}\|\mathbf{w}-\mathbf{1}/K\|_2^2$ in problem (4), we then perform one-step PGD (w.r.t. $-\psi$) to update $\mathbf{w}$,

$$\mathbf{w}^{(t)} = \text{proj}_{\mathcal{P}}\Big(\underbrace{\mathbf{w}^{(t-1)} + \beta \nabla_{\mathbf{w}}\psi(\mathbf{w}^{(t-1)})}_{\mathbf{b}}\Big) = (\mathbf{b} - \mu\mathbf{1})_+, \qquad (9)$$

where $\beta > 0$ is a given learning rate, $\nabla_{\mathbf{w}}\psi(\mathbf{w}) = \boldsymbol{\phi}^{(t)} - \gamma(\mathbf{w} - \mathbf{1}/K)$, and $\boldsymbol{\phi}^{(t)} := [F_1(\boldsymbol{\delta}^{(t)}), \ldots, F_K(\boldsymbol{\delta}^{(t)})]^T$. In (9), the second equality holds due to the closed-form of projection operation onto the probabilistic simplex $\mathcal{P}$ [49], where $(x)_+ = \max\{0, x\}$, and $\mu$ is the root of the equation $\mathbf{1}^T(\mathbf{b} - \mu\mathbf{1})_+ = 1$. Since $\mathbf{1}^T(\mathbf{b} - \min_i\{b_i\}\mathbf{1} + \mathbf{1}/K)_+ \geq \mathbf{1}^T\mathbf{1}/K = 1$, and $\mathbf{1}^T(\mathbf{b} - \max_i\{b_i\}\mathbf{1} + \mathbf{1}/K)_+ \leq \mathbf{1}^T\mathbf{1}/K = 1$, the root $\mu$ exists within the interval $[\min_i\{b_i\} - 1/K, \max_i\{b_i\} - 1/K]$ and can be found via the bisection method [8].

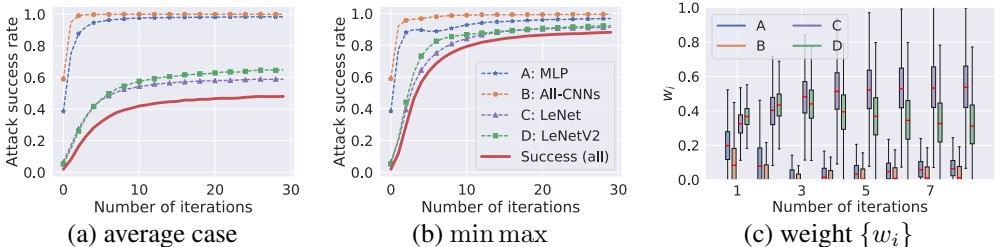

|(a) average case|(b) min max|(c) weight $\{w_i\}$|

Figure 1: Ensemble attack against four DNN models on MNIST. (a) & (b): Attack success rate of adversarial examples generated by average PGD or min-max (APGDA) attack method. (c): Boxplot of weight $w$ in min-max adversarial loss. Here we adopt the same $\ell_\infty$-attack as Table 1.

**Convergence Analysis** We remark that APGDA follows the gradient primal-dual optimization framework [37], and thus enjoys the same optimization guarantees.

**Theorem 1.** *Suppose that in problem (4) $F_i(\boldsymbol{\delta})$ has L-Lipschitz continuous gradients, and $\mathcal{X}$ is a convex compact set. Given learning rates $\alpha \leq \frac{1}{L}$ and $\beta < \frac{1}{\gamma}$, then the sequence $\{\boldsymbol{\delta}^{(t)}, \mathbf{w}^{(t)}\}_{t=1}^T$ generated by Algorithm 1 converges to a first-order stationary point[2] in rate $\mathcal{O}\left(\frac{1}{T}\right)$.*

*Proof*: Note that the objective function of problem (4) is strongly concave w.r.t. $\mathbf{w}$ with parameter $\gamma$, and has $\gamma$-Lipschitz continuous gradients. Moreover, we have $\|\mathbf{w}\|_2 \leq 1$ due to $\mathbf{w} \in \mathcal{P}$. Using these facts and Theorem 1 in [37] or [39] completes the proof. □

## 4 Experiments on Adversarial Exploration

In this section, we first evaluate the proposed min-max optimization strategy on three attack tasks. We show that our approach leads to substantial improvement compared with state-of-the-art attack methods such as average ensemble PGD [34] and EOT [3, 10, 5]. We also demonstrate the effectiveness of learnable domain weights in guiding the adversary's exploration over multiple domains.

### 4.1 Experimental setup

We thoroughly evaluate our algorithm on MNIST and CIFAR-10. A set of diverse image classifiers (denoted from Model A to Model H) are trained, including multi-layer perceptron (MLP), All-CNNs [61], LeNet [30], LeNetV2, VGG16 [58], ResNet50 [24], Wide-ResNet [40, 75] and GoogLeNet [63]. The details about model architectures and training process are provided in Appendix D.1. Note that problem formulations (5)-(7) are applicable to both *untargeted* and *targeted* attack. Here we focus on the former setting and use C&W loss function [13, 40] with a confidence parameter $\kappa = 50$. The adversarial examples are generated by 20-step PGD/APGDA unless otherwise stated (e.g., 50 steps for ensemble attacks). APGDA algorithm is relatively robust and will not be affected largely by the choices of hyperparameters $(\alpha, \beta, \gamma)$. Apart from absolute attack success rate (ASR), we also report the relative improvement or degradationon the worse-case performance in experiments: Lift($\uparrow$). The details of crafting adversarial examples are available in Appendix D.2.

### 4.2 Ensemble Attack over Multiple Models

We craft adversarial examples against an ensemble of known classifiers. Recent work [34] proposed an average ensemble PGD attack, which assumed equal importance among different models, namely, $w_i = 1/K$ in problem (5). Throughout this task, we measure the attack performance via **ASR$_{all}$** - the attack success rate (ASR) of fooling model ensembles simultaneously. Compared to the average PGD attack, our approach results in 40.79% and 17.48% ASR$_{all}$ improvement averaged over different $\ell_p$-norm constraints on MNIST and CIFAR-10, respectively. In what follows, we provide more detailed results and analysis.

In Table 1 and Table 3, we show that AMGDA significantly outperforms average PGD in ASR$_{all}$. Taking $\ell_\infty$-attack on MNIST as an example, our min-max attack leads to a 90.16% ASR$_{all}$, which

---

[2]The stationarity is measured by the $\ell_2$ norm of gradient of the objective in (4) w.r.t. $(\boldsymbol{\delta}, \mathbf{w})$.

Table 1: Comparison of average and min-max (APGDA) ensemble attack on MNIST.

| Box constraint | Opt. | $Acc_A$ | $Acc_B$ | $Acc_C$ | $Acc_D$ | $ASR_{all}$ | Lift (↑) |
|---|---|---|---|---|---|---|---|
| $\ell_0$ ($\epsilon = 30$) | avg. | 7.03 | 1.51 | 11.27 | 2.48 | 84.03 | - |
|  | min max | 3.65 | 2.36 | 4.99 | 3.11 | **91.97** | **9.45%** |
| $\ell_1$ ($\epsilon = 20$) | avg. | 20.79 | 0.15 | 21.48 | 6.70 | 69.31 | - |
|  | min max | 6.12 | 2.53 | 8.43 | 5.11 | **89.16** | **28.64%** |
| $\ell_2$ ($\epsilon = 3.0$) | avg. | 6.88 | 0.03 | 26.28 | 14.50 | 69.12 | - |
|  | min max | 1.51 | 0.89 | 3.50 | 2.06 | **95.31** | **37.89%** |
| $\ell_\infty$ ($\epsilon = 0.2$) | avg. | 1.05 | 0.07 | 41.10 | 35.03 | 48.17 | - |
|  | min max | 2.47 | 0.37 | 7.39 | 5.81 | **90.16** | **87.17%** |

Table 2: Comparison to heuristic weighting schemes on MNIST ($\ell_\infty$-attack, $\epsilon = 0.2$).

| Opt. | $Acc_A$ | $Acc_B$ | $Acc_C$ | $Acc_D$ | $ASR_{avg}$ | $ASR_{all}$ | Lift (↑) |
|---|---|---|---|---|---|---|---|
| avg. | 1.05 | 0.07 | 41.10 | 35.03 | 80.69 | 48.17 | - |
| $w_{c+d}$ | 60.37 | 19.55 | 15.10 | 1.87 | 75.78 | 29.32 | -39.13% |
| $w_{a+c+d}$ | 0.46 | 21.57 | 25.36 | 13.84 | 84.69 | 53.39 | 10.84% |
| $w_{clip}$ [56] | 0.66 | 0.03 | 23.43 | 13.23 | 90.66 | 71.54 | 48.52% |
| $w_{prior}$ | 1.57 | 0.24 | 17.67 | 13.74 | 91.70 | 74.34 | 54.33% |
| $w_{static}$ | 10.58 | 0.39 | 9.28 | 10.05 | 92.43 | 77.84 | 61.59% |
| min max | 2.47 | 0.37 | 7.39 | 5.81 | **95.99** | **90.16** | **87.17%** |

Table 3: Comparison of average and min-max (APGDA) ensemble attack on CIFAR-10.

| Box constraint | Opt. | $Acc_A$ | $Acc_B$ | $Acc_C$ | $Acc_D$ | $ASR_{all}$ | Lift (↑) |
|---|---|---|---|---|---|---|---|
| $\ell_0$ ($\epsilon = 50$) | avg. | 27.86 | 3.15 | 5.16 | 6.17 | 65.16 | - |
|  | min max | 18.74 | 8.66 | 9.64 | 9.70 | **71.44** | **9.64%** |
| $\ell_1$ ($\epsilon = 30$) | avg. | 32.92 | 2.07 | 5.55 | 6.36 | 59.74 | - |
|  | min max | 12.46 | 3.74 | 5.62 | 5.86 | **78.65** | **31.65%** |
| $\ell_2$ ($\epsilon = 2.0$) | avg. | 24.3 | 1.51 | 4.59 | 4.20 | 69.55 | - |
|  | min max | 7.17 | 3.03 | 4.65 | 5.14 | **83.95** | **20.70%** |
| $\ell_\infty$ ($\epsilon = 0.05$) | avg. | 19.69 | 1.55 | 5.61 | 4.26 | 73.29 | - |
|  | min max | 7.21 | 2.68 | 4.74 | 4.59 | **84.36** | **15.10%** |

Table 4: Comparison to heuristic weighting schemes on CIFAR-10 ($\ell_\infty$-attack, $\epsilon = 0.05$).

| Opt. | $Acc_A$ | $Acc_B$ | $Acc_C$ | $Acc_D$ | $ASR_{avg}$ | $ASR_{all}$ | Lift (↑) |
|---|---|---|---|---|---|---|---|
| avg. | 19.69 | 1.55 | 5.61 | 4.26 | 92.22 | 73.29 | - |
| $w_{b+c+d}$ | 42.12 | 1.63 | 5.93 | 4.42 | 75.78 | 51.63 | -29.55% |
| $w_{a+c+d}$ | 13.33 | 32.41 | 4.83 | 5.44 | 84.69 | 56.89 | -22.38% |
| $w_{clip}$ [56] | 11.13 | 3.75 | 6.66 | 6.02 | 90.66 | 77.82 | 6.18% |
| $w_{prior}$ | 19.72 | 2.30 | 4.38 | 4.29 | 91.70 | 73.45 | 0.22% |
| $w_{static}$ | 7.36 | 4.48 | 5.03 | 6.70 | 92.43 | 81.04 | 10.57% |
| min max | 7.21 | 2.68 | 4.74 | 4.59 | **95.20** | **84.36** | **15.10%** |

largely outperforms 48.17%. The reason is that Model C, D are more difficult to attack, which can be observed from their higher test accuracy on adversarial examples. As a result, although the adversarial examples crafted by assigning equal weights over multiple models are able to attack {A, B} well, they achieve a much lower ASR in {C, D}. By contrast, APGDA automatically handles the worst case {C, D} by slightly sacrificing the performance on {A, B}: 31.47% averaged ASR improvement on {C, D} versus 0.86% degradation on {A, B}. The choices of $\alpha, \beta, \gamma$ for all experiments and more results on CIFAR-10 are provided in Appendix D.2 and Appendix E.

**Effectiveness of learnable domain weights:**
Figure 1 depicts the ASR of four models under average/min-max attacks as well as the distribution of domain weights during attack generation. For average PGD (Figure 1a), Model C and D are attacked insufficiently, leading to relatively low ASR and thus weak ensemble performance. By contrast, APGDA (Figure 1b) will encode the difficulty level to attack different models based on the current attack loss. It dynamically adjusts the weight $w_i$ as shown in Figure 1c. For

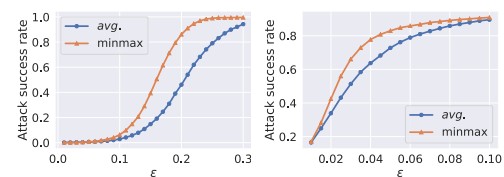

Figure 2: ASR of average and min-max $\ell_\infty$ ensemble attack versus maximum perturbation magnitude $\epsilon$. Left (MNIST), Right (CIFAR-10).

instance, the weight for Model D is first raised to $0.45$ because D is difficult to attack initially. Then it decreases to $0.3$ once Model D encounters the sufficient attack power and the corresponding attack performance is no longer improved. It is worth noticing that APGDA is highly efficient because $w_i$ converges after a small number of iterations. Figure 1c also shows $w_c > w_d > w_a > w_b$ – indicating a decrease in model robustness for C, D, A and B, which is exactly verified by $Acc_C > Acc_D > Acc_A > Acc_B$ in the last row of Table 1 ($\ell_\infty$-norm). As the perturbation radius $\epsilon$ varies, we also observe that the ASR of min-max strategy is consistently better or on part with the average strategy (see Figure 2).

**Comparison with stronger heuristic baselines** Apart from *average* strategy, we compare min-max framework with stronger heuristic weighting scheme in Table 2 (MNIST) and Table 4 (CIFAR-10). Specifically, with the prior knowledge of robustness of given models ($C > D > A > B$), we devised several heuristic baselines including: (a) $w_{c+d}$: average PGD on models C and D only; (b) $w_{a+c+d}$: average PGD on models A, C and D only; (c) $w_{clip}$: clipped version of C&W loss (threshold $\beta = 40$) to balance model weights in optimization as suggested in [56]; (d) $w_{prior}$: larger weights on the more robust models, $w_{prior} = [w_A, w_B, w_C, w_D] = [0.2, 0.1, 0.4, 0.3]$; (e) $w_{static}$: the converged mean weights of min-max (APGDA) ensemble attack. For $\ell_2$ ($\epsilon = 3.0$) and $\ell_\infty$ ($\epsilon = 0.2$) attacks, $w_{static} = [w_A, w_B, w_C, w_D]$ are $[0.209, 0.046, 0.495, 0.250]$ and $[0.080, 0.076, 0.541, 0.303]$, respectively. Table 2 shows that our approach achieve substantial improvement over baselines consistently. Moreover, we highlight that the use of learnable **w** avoids supervised manual adjustment on

Table 5: Comparison of average and minmax optimization on universal perturbation over multiple input examples. $K$ represents the number of images in each group. $\text{ASR}_{avg}$ and $\text{ASR}_{all}$ mean attack success rate (%) of all images and success rate of attacking all the images in each group, respectively. The adversarial examples are generated by 20-step $\ell_\infty$-APGDA with $\alpha = \frac{1}{6}, \beta = \frac{1}{50}$ and $\gamma = 4$.

| Dataset | Model | Opt. | $\text{ASR}_{avg}$ | K=2 $\text{ASR}_{all}$ | Lift (↑) | $\text{ASR}_{avg}$ | K=4 $\text{ASR}_{all}$ | Lift (↑) | $\text{ASR}_{avg}$ | K=5 $\text{ASR}_{all}$ | Lift (↑) | $\text{ASR}_{avg}$ | K=10 $\text{ASR}_{all}$ | Lift (↑) |
|---|---|---|---|---|---|---|---|---|---|---|---|---|---|---|
| CIFAR-10 | All-CNNs | avg. | 91.09 | 83.08 | - | 85.66 | 54.72 | - | 82.76 | 40.20 | - | 71.22 | 4.50 | - |
| | | min max | 92.22 | **85.98** | **3.49%** | 87.63 | **65.80** | **20.25%** | 85.02 | **55.74** | **38.66%** | 65.64 | **11.80** | **162.2%** |
| | LeNetV2 | avg. | 93.26 | 86.90 | - | 90.04 | 66.12 | - | 88.28 | 55.00 | - | 72.02 | 8.90 | - |
| | | min max | 93.34 | **87.08** | **0.21%** | 91.91 | **71.64** | **8.35%** | 91.21 | **63.55** | **15.55%** | 82.85 | **25.10** | **182.0%** |
| | VGG16 | avg. | 90.76 | 82.56 | - | 89.36 | 63.92 | - | 88.74 | 55.20 | - | 85.86 | 22.40 | - |
| | | min max | 92.40 | **85.92** | **4.07%** | 90.04 | **70.40** | **10.14%** | 88.97 | **63.30** | **14.67%** | 79.07 | **30.80** | **37.50%** |
| | GoogLeNet | avg. | 85.02 | 72.48 | - | 75.20 | 32.68 | - | 71.82 | 19.60 | - | 59.01 | 0.40 | - |
| | | min max | 87.08 | **77.82** | **7.37%** | 77.05 | **46.20** | **41.37%** | 71.20 | **33.70** | **71.94%** | 45.46 | **2.40** | **600.0%** |

Table 6: Interpretability of domain weight $w$ for universal perturbation to multiple inputs on MNIST (*Digit 0, 2, 4*). Domain weight $w$ for different images under $\ell_p$-norm ($p = 0, 1, 2, \infty$).

| | Image | | | | | | | | | | | | | | |
|---|---|---|---|---|---|---|---|---|---|---|---|---|---|---|---|
| Weight | $\ell_0$ | 0. | 0. | 0. | 0. | 1.000 | 0. | 0. | 0.909 | 0. | 0.091 | 0. | 0. | 0.753 | 0. | 0.247 |
| | $\ell_1$ | 0. | 0. | 0. | 0. | 1.000 | 0. | 0. | 0.843 | 0. | 0.157 | 0.018 | 0. | 0.567 | 0. | 0.416 |
| | $\ell_2$ | 0. | 0. | 0. | 0. | 1.000 | 0. | 0. | 0.788 | 0. | 0.112 | 0. | 0. | 0.595 | 0. | 0.405 |
| | $\ell_\infty$ | 0. | 0. | 0. | 0. | 1.000 | 0. | 0. | 0.850 | 0. | 0.150 | 0. | 0. | 0.651 | 0. | 0.349 |
| Metric | dist.(C&W $\ell_2$) | 1.839 | 1.954 | 1.347 | 1.698 | **3.041** | 1.928 | 1.439 | 2.312 | 1.521 | **2.356** | 1.558 | 1.229 | **1.939** | 0.297 | 1.303 |
| | $\epsilon_{\min}$ ($\ell_\infty$) | 0.113 | 0.167 | 0.073 | 0.121 | **0.199** | 0.082 | 0.106 | **0.176** | 0.072 | 0.171 | 0.084 | 0.088 | **0.122** | 0.060 | 0.094 |

the heuristic weights or the choice of clipping threshold. Also, we show that even adopting converged min-max weights statically leads to a huge performance drop on attacking model ensembles, which again verifies the power of dynamically optimizing domain weights during attack generation process.

## 4.3 Multi-Image Universal Perturbation

We evaluate APGDA in universal perturbation on MNIST and CIFAR-10, where 10,000 test images are randomly divided into equal-size groups ($K$ images per group) for universal perturbation. We measure two types of ASR (%), **$\text{ASR}_{avg}$** and **$\text{ASR}_{all}$**. Here the former represents the ASR averaged over all images in all groups, and the latter signifies the ASR averaged over all groups but a successful attack is counted under a more restricted condition: images within each group must be successfully attacked simultaneously by universal perturbation. In Table 5, we compare the proposed min-max strategy with the averaging strategy on the attack performance of generated universal perturbations. APGDA always achieves higher $\text{ASR}_{all}$ for different values of $K$. When $K = 5$, our approach achieves 42.63% and 35.21% improvement over the averaging strategy under MNIST and CIFAR-10. The universal perturbation generated from APGDA can successfully attack 'hard' images (on which the average-based PGD attack fails) by self-adjusting domain weights, and thus leads to a higher $\text{ASR}_{all}$.

**Interpreting "*image robustness*" with domain weights w:** The min-max universal perturbation also offers interpretability of "*image robustness*" by associating domain weights with image visualization. Figure 6 shows an example in which the large domain weight corresponds to the MNIST letter with clear appearance (e.g., bold letter). To empirically verify the robustness of image, we report two metrics to measure the difficulty of attacking single image: dist. (C&W $\ell_2$) denotes the the minimum distortion of successfully attacking images using C&W ($\ell_2$) attack; $\epsilon_{\min}$ ($\ell_\infty$) denotes the minimum perturbation magnitude for $\ell_\infty$-PGD attack.

## 4.4 Robust Attack over Data Transformations

EOT [5] achieves state-of-the-art performance in producing adversarial examples robust to data transformations. From (7), we could derive EOT as a special case when the weights satisfy $w_i = 1/K$ (average case). For each input sample (*ori*), we transform the image under a series of functions, e.g., flipping horizontally (*flh*) or vertically (*flv*), adjusting brightness (*bri*), performing gamma correction

Table 7: Comparison of average and min-max optimization on robust attack over multiple data transformations on CIFAR-10. Acc (%) represents the test accuracy of classifiers on adversarial examples (20-step $\ell_\infty$-APGD ($\epsilon = 0.03$) with $\alpha = \frac{1}{2}, \beta = \frac{1}{100}$ and $\gamma = 10$) under different transformations.

| Model | Opt. | $\text{Acc}_{ori}$ | $\text{Acc}_{flh}$ | $\text{Acc}_{flv}$ | $\text{Acc}_{bri}$ | $\text{Acc}_{gam}$ | $\text{Acc}_{crop}$ | $\text{ASR}_{all}$ | Lift ($\uparrow$) |
|---|---|---|---|---|---|---|---|---|---|
| A | *avg.* | 10.80 | 21.93 | 14.75 | 11.52 | 10.66 | 20.03 | 55.88 | - |
|   | min max | 12.14 | 18.05 | 13.61 | 13.52 | 11.99 | 16.78 | 60.03 | **7.43%** |
| B | *avg.* | 5.49 | 11.56 | 9.51 | 5.43 | 5.75 | 15.89 | 72.21 | - |
|   | min max | 6.22 | 8.61 | 9.74 | 6.35 | 6.42 | 11.99 | 77.43 | **7.23%** |
| C | *avg.* | 7.66 | 21.88 | 15.50 | 8.15 | 7.87 | 15.36 | 56.51 | - |
|   | min max | 8.51 | 14.75 | 13.88 | 9.16 | 8.58 | 13.35 | 63.58 | **12.51%** |
| D | *avg.* | 8.00 | 20.47 | 13.46 | 7.73 | 8.52 | 15.90 | 61.13 | - |
|   | min max | 9.19 | 13.18 | 12.72 | 8.79 | 9.18 | 13.11 | **67.49** | **10.40%** |

(*gam*) and cropping (*crop*), and group each image with its transformed variants. Similar to universal perturbation, $\text{ASR}_{all}$ is reported to measure the ASR over groups of transformed images (each group is successfully attacked signifies successfully attacking an example under all transformers). In Table 7, compared to EOT, our approach leads to 9.39% averaged lift in $\text{ASR}_{all}$ over given models on CIFAR-10 by optimizing the weights for various transformations. We leave the the results under randomness (e.g., flipping images randomly *w.p.* 0.8; randomly clipping the images at specific range) in Appendix E

## 5 Extension: Understanding Defense over Multiple Perturbation Domains

In this section, we show that the min-max principle can also be used to gain more insights in generalized adversarial training (AT) from a defender's perspective. Different from promoting robustness of adversarial examples against the *worst-case attacking environment* (Sec. 3), the generalized AT promotes model's robustness against the *worst-case defending environment*, given by the existence of multiple $\ell_p$ attacks [65]. Our approach obtains better performance than prior works [65, 41] and interpretability by introducing the trainable domain weights.

### 5.1 Adversarial Training under Mixed Types of Adversarial Attacks

Conventional AT is restricted to a single type of norm-ball constrained adversarial attack [40]. For example, AT under $\ell_\infty$ attack yields:

$$\underset{\boldsymbol{\theta}}{\text{minimize}} \; \mathbb{E}_{(\mathbf{x}, \mathbf{y}) \in \mathcal{D}} \underset{\|\boldsymbol{\delta}\|_\infty \leq \epsilon}{\text{maximize}} \; f_{\text{tr}}(\boldsymbol{\theta}, \boldsymbol{\delta}; \mathbf{x}, y), \tag{10}$$

where $\boldsymbol{\theta} \in \mathbb{R}^n$ denotes model parameters, $\boldsymbol{\delta}$ denotes $\epsilon$-tolerant $\ell_\infty$ attack, and $f_{\text{tr}}(\boldsymbol{\theta}, \boldsymbol{\delta}; \mathbf{x}, y)$ is the training loss under perturbed examples $\{(\mathbf{x} + \boldsymbol{\delta}, y)\}$. However, there possibly exist blind attacking spots across multiple types of adversarial attacks so that AT under one attack would not be strong enough against another attack [2]. Thus, an interesting question is how to generalize AT under multiple types of adversarial attacks [65]. One possible way is to use the finite-sum formulation in the inner maximization problem of (10), namely, $\text{maximize}_{\{\boldsymbol{\delta}_i \in \mathcal{X}_i\}} \frac{1}{K} \sum_{i=1}^{K} f_{\text{tr}}(\boldsymbol{\theta}, \boldsymbol{\delta}_i; \mathbf{x}, y)$, where $\boldsymbol{\delta}_i \in \mathcal{X}_i$ is the $i$th type of adversarial perturbation defined on $\mathcal{X}_i$, e.g., different $\ell_p$ attacks.

Since we can map 'attack type' to 'domain' considered in (1), AT can be generalized against the *strongest* adversarial attack across $K$ attack types in order to avoid blind attacking spots:

$$\underset{\boldsymbol{\theta}}{\text{minimize}} \; \mathbb{E}_{(\mathbf{x}, \mathbf{y}) \in \mathcal{D}} \underset{i \in [K]}{\text{maximize}} \underset{\boldsymbol{\delta}_i \in \mathcal{X}_i}{\text{maximize}} \; f_{\text{tr}}(\boldsymbol{\theta}, \boldsymbol{\delta}_i; \mathbf{x}, y). \tag{11}$$

In Lemma 1, we show that problem (11) can be equivalently transformed into the min-max form.

**Lemma 1.** *Problem (11) is equivalent to:*

$$\underset{\boldsymbol{\theta}}{\text{minimize}} \; \mathbb{E}_{(\mathbf{x}, \mathbf{y}) \in \mathcal{D}} \underset{\mathbf{w} \in \mathcal{P}, \{\boldsymbol{\delta}_i \in \mathcal{X}_i\}}{\text{maximize}} \sum_{i=1}^{K} w_i f_{\text{tr}}(\boldsymbol{\theta}, \boldsymbol{\delta}_i; \mathbf{x}, y), \tag{12}$$

*where $\mathbf{w} \in \mathbb{R}^K$ represent domain weights, and $\mathcal{P}$ has been defined in (1).*

| | MAX [3] | AVG [3] | MSD [2] | AMPGD |
|---|---|---|---|---|
| Clean Accuracy | 98.6% | 99.1% | 98.3% | 98.3% |
| $\ell_\infty$ Attacks [65] ($\epsilon = 0.3$) | 51.0% | 65.2% | 62.7% | 76.1% |
| $\ell_2$ Attacks [65] ($\epsilon = 2.0$) | 61.9% | 60.1% | 67.9% | 70.2% |
| $\ell_1$ Attacks [65] ($\epsilon = 10$) | 52.6% | 39.2% | 65.0% | 67.2% |
| All Attacks [65] | 42.1% | 34.9% | 58.4% | **64.1%** |
| AA (all attacks) [18] | 36.9% | 30.5% | 55.9% | 59.3% |
| AA+ (all attacks) [18] | 34.3% | 28.8% | 54.8% | **58.3%** |

Table 8: Adversarial robustness on MNIST.

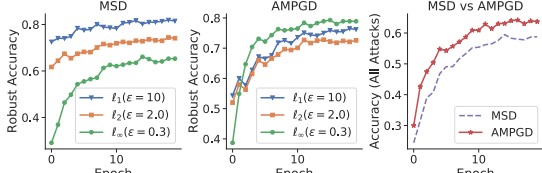

Figure 3: Robust accuracy of MSD and AMPGD.

| | $L_\infty$-AT | $L_2$-AT | $L_1$-AT | MAX [65] | AVG [66] | MSD [41] | AMPGD |
|---|---|---|---|---|---|---|---|
| Clean Accuracy | 83.3% | 90.2% | 73.3% | 81.0% | 84.6% | 81.1% | 81.5% |
| $\ell_\infty$ Attacks ($\epsilon = 0.03$) [41] | 50.7% | 28.3% | 0.2% | 44.9% | 42.5% | 48.0% | 49.2% |
| $\ell_2$ Attacks ($\epsilon = 0.5$) [41] | 57.3% | 61.6% | 0.0% | 61.7% | 65.0% | 64.3% | 68.0% |
| $\ell_1$ Attacks ($\epsilon = 12$) [41] | 16.0% | 46.6% | 7.9% | 39.4% | 54.0% | 53.0% | 50.0% |
| All Attacks [41] | 15.6% | 27.5% | 0.0% | 34.9% | 40.6% | 47.0% | **48.7%** |
| AA ($\ell_\infty, \epsilon = 0.03$) [18] | 47.8% | 22.7% | 0.0% | 39.2% | 40.7% | 44.4% | 46.9% |
| AA ($\ell_2, \epsilon = 0.5$) [18] | 57.5% | 63.1% | 0.1% | 62.0% | 65.5% | 64.9% | 64.4% |
| AA ($\ell_1, \epsilon = 12$) [18] | 13.7% | 23.6% | 1.4% | 36.0% | 58.8% | 52.4% | 52.3% |
| AA (all attacks) [18] | 12.8% | 18.4% | 0.0% | 30.8% | 40.4% | 44.1% | **46.2%** |

Table 9: Summary of adversarial accuracy results for CIFAR-10.

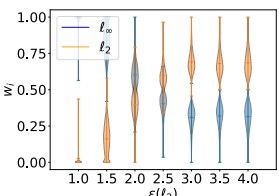

Figure 4: Domain weights.

The proof of Lemma 1 is provided in Appendix B. Similar to (4), a strongly concave regularizer $-\gamma/2\|\mathbf{w} - 1/K\|_2^2$ can be added into the inner maximization problem of (12) for boosting the stability of the learning procedure and striking a balance between the max and the average attack performance:

$$\begin{aligned} \underset{\boldsymbol{\theta}}{\text{minimize}} \ \mathbb{E}_{(\mathbf{x},\mathbf{y})\in\mathcal{D}} \ \underset{\mathbf{w}\in\mathcal{P},\{\boldsymbol{\delta}_i\in\mathcal{X}_i\}}{\text{maximize}} \ \psi(\boldsymbol{\theta},\mathbf{w},\{\boldsymbol{\delta}_i\}) \\ \psi(\boldsymbol{\theta},\mathbf{w},\{\boldsymbol{\delta}_i\}) := \sum_{i=1}^K w_i f_{\text{tr}}(\boldsymbol{\theta},\boldsymbol{\delta}_i;\mathbf{x},y) - \frac{\gamma}{2}\|\mathbf{w} - 1/K\|_2^2 \end{aligned} \tag{13}$$

We propose the **a**lternating **m**ulti-step **p**rojected **g**radient **d**escent (AMPGD) method (Algorithm 2) to solve problem (13). Since AMPGD also follows the min-max principles, we defer more details of this algorithm in Appendix C. We finally remark that our formulation of generalized AT under multiple perturbations covers prior work [65] as special cases ($\gamma = 0$ for max case and $\gamma = \infty$ for average case).

---

**Algorithm 2** AMPGD to solve problem (13)

1: Input: given $\boldsymbol{\theta}^{(0)}, \mathbf{w}^{(0)}, \boldsymbol{\delta}^{(0)}$ and $K > 0$.
2: **for** $t = 1, 2, \ldots, T$ **do**
3:     given $\mathbf{w}^{(t-1)}$ and $\boldsymbol{\delta}^{(t-1)}$, perform SGD to update $\boldsymbol{\theta}^{(t)}$
4:     given $\boldsymbol{\theta}^{(t)}$, perform $R$-step PGD to update $\mathbf{w}^{(t)}$ and $\boldsymbol{\delta}^{(t)}$
5: **end for**

---

### 5.2 Generalized AT vs. Multiple $\ell_p$ Attacks

Compared to vanilla AT, we show the generalized AT scheme produces model robust to multiple types of perturbation, thus leads to stronger "overall robustness". We present experimental results of generalized AT following [41] to achieve simultaneous robustness to $\ell_\infty$, $\ell_2$, and $\ell_1$ perturbations on the MNIST and CIFAR-10 datasets. To the best of our knowledge, MSD proposed in [41] is the state-of-the-art defense against multiple types of $\ell_p$ attacks. Specifically, we adopted the same architectures as [41] four layer convolutional networks on MNIST and the pre-activation version of the ResNet18 [24]. The perturbation radius $\epsilon$ for ($\ell_\infty, \ell_2, \ell_1$) balls is set as $(0.3, 2.0, 10)$ and $(0.03, 0.5, 12)$ on MNIST and CIFAR-10 following [41]. Apart from the evaluation $\ell_p$ PGD attacks, we also incorporate the state-of-the-art AutoAttack [18] for a more comprehensive evaluation under mixed $\ell_p$ perturbations.

The adversarial accuracy results are reported (higher the better). As shown in Table 8 and 9, our approach outperforms the state-of-the-art defense MSD consistently (4∼6% and 2% improvements on MNIST and CIFAR-10). Compared to MSD that deploys an approximate arg max operation to select the steepest-descent (worst-case) universal perturbation, we leverage the domain weights to self-adjust the strengthens of diverse $\ell_p$ attacks. Thus, we believe that this helps gain supplementary robustness from individual attacks.

**Effectiveness of Domain Weights:** Figure 3 shows the robust accuracy curves of MSD and AMPGD on MNIST. As we can see, the proposed AMPGD can quickly adjust the defense strengths to focus on more difficult adversaries - the gap of robust accuracy between three attacks is much smaller. Therefore, it achieves better results by avoiding the trade-off that biases one particular perturbation model at the cost of the others. In Figure 4, we offer deeper insights on how the domain weights work as the strengths of adversary vary. Specifically, we consider two perturbation models on MNIST: $\ell_2$ and $\ell_\infty$. During the training, we fix the $\epsilon$ for $\ell_\infty$ attack during training as 0.2, and change the $\epsilon$ for $\ell_2$ from 1.0 to 4.0. As shown in Figure 4, the domain weight $w$ increases when the $\ell_2$-attack becomes stronger i.e., $\epsilon(\ell_2)$ increases, which is consistent with min-max spirit – defending the strongest attack.

### 5.3 Additional Discussions

**More parameters to tune for min-max?** Our min-max approaches (APGDA and AMPGD) introduce two more hyperparameters - $\beta$ and $\gamma$. However, our proposal performs reasonably well by choosing the learning rate $\alpha$ same as standard PGD and using a large range of regularization coefficient $\gamma \in [0, 10]$; see Fig. A5 in Appendix. For the learning rate $\beta$ to update domain weights, we found $1/T$ is usually a very good practice, where $T$ is the total number of attack iterations.

**Time complexity of inner maximization?** Our proposal achieves significant improvements at a low cost of extra computation. Specifically, (1) our APGDA attack is $1.31\times$ slower than the average PGD; (2) our AMPGD defense is $1.15\times$ slower than average or max AT [65].

**How efficient is the APGDA (Algorithm 1) for solving problem (4)?** We remark that the min-max attack generation setup obeys the nonconvex + strongly concave optimization form. Our proposed APGDA is a single-loop algorithm, which is known to achieve a nearly optimal convergence rate for nonconvex-strongly concave min-max optimization [32, Table 1]. Furthermore, as our solution gives a natural extension from the commonly-used PGD attack algorithm by incorporating the inner maximization step (9), it is easy to implement based on existing frameworks.

**Clarification on contributions:** Our contribution is not to propose a new or more efficient optimization approach for solving min-max optimization problems. Instead, we focus on introducing this formulation to the attack design domain, which has not been studied systematically before. We believe this work is the first solid step to explore the power of min-max principle in the attack design and achieve superior performance on multiple attack tasks.

## 6 Conclusion

In this paper, we revisit the strength of min-max optimization in the context of adversarial attack generation. Beyond adversarial training (AT), we show that many attack generation problems can be re-formulated in our unified min-max framework, where the maximization is taken over the probability simplex of the set of domains. Experiments show our min-max attack leads to significant improvements on three tasks. Importantly, we demonstrate the self-adjusted domain weights not only stabilize the training procedure but also provides a holistic tool to interpret the risk of different domain sources. Our min-max principle also helps understand the generalized AT against multiple adversarial attacks. Our approach results in superior performance as well as intepretability.

## Broader Impacts

Our work provides a unified framework in design of adversarial examples and robust defenses. The generated adversarial examples can be used to evaluate the robustness of state-of-the-art deep learning vision systems. In spite of different kinds of adversaries, the proposed defense solves one for all by taking into account adversaries' diversity. Our work is a beneficial supplement to building trustworthy AI systems, in particular for safety-critical AI applications, such as autonomous vehicles and camera surveillance. We do not see negative impacts of our work on its ethical aspects and future societal consequences.

## Acknowledgement

We sincerely thank the anonymous reviewers for their insightful suggestions and feedback. This work is partially supported by the NSF grant No.1910100, NSF CNS 20-46726 CAR, NSF CAREER CMMI-1750531, NSF ECCS-1609916, and the Amazon Research Award. Resources used in preparing this research were provided, in part, by the Province of Ontario, the Government of Canada through CIFAR, and companies sponsoring the Vector Institute.

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
