# Supplementary Material
## Adversarial Attack Generation Empowered by Min-Max Optimization

## Abstract

In this supplementary material, we first provide technical proofs of Proposition 1 and Lemma 1 in Sec A and B. We then discuss the proposed AMPGD algorithm in Sec C. In the next section, we show the details of experimental setup including the model architectures and training details in Sec D.1, the hyperparameters to craft the adversarial examples (Sec D.2), the details of data transformations (Sec D.3). Then we show additional experiments results for robust adversarial attacks (Sec E) and generalized adversarial training (Sec F). Finally, we provide more visualizations to show that domain weights $\mathbf{w}$ provide a holistic tool to interpret "image robustness" in Sec G. The summary of contents in the supplementary is provided in the following.

## Contents

# A  Proof of Proposition 1

**Proposition 1.** *Given a point* $\mathbf{a} \in \mathbb{R}^d$ *and a constraint set* $\mathcal{X} = \{\boldsymbol{\delta} | \|\boldsymbol{\delta}\|_p \leq \epsilon, \check{\mathbf{c}} \leq \boldsymbol{\delta} \leq \hat{\mathbf{c}}\}$, *the Euclidean projection* $\boldsymbol{\delta}^* = \text{proj}_{\mathcal{X}}(\mathbf{a})$ *has the closed-form solution when* $p \in \{0, 1, 2\}$.

*1) If* $p = 1$, *then* $\boldsymbol{\delta}^*$ *is given by*

$$\delta_i^* = \begin{cases} P_{[\check{c}_i, \hat{c}_i]}(a_i) & \sum_{i=1}^d |P_{[\check{c}_i, \hat{c}_i]}(a_i)| \leq \epsilon \\ P_{[\check{c}_i, \hat{c}_i]}(\text{sign}(a_i) \max\{|a_i| - \lambda_1, 0\}) & \text{otherwise}, \end{cases} \tag{14}$$

*where* $\mathbf{x}_i$ *denotes the* $i$*th element of a vector* $\mathbf{x}$; $P_{[\check{c}_i, \hat{c}_i]}(\cdot)$ *denotes the clip function over the interval* $[\check{c}_i, \hat{c}_i]$; $\text{sign}(x) = 1$ *if* $x \geq 0$, *otherwise* $0$; $\lambda_1 \in (0, \max_i |a_i| - \epsilon/d]$ *is the root of* $\sum_{i=1}^d |P_{[\check{c}_i, \hat{c}_i]}(\text{sign}(a_i) \max\{|a_i| - \lambda_1, 0\})| = \epsilon$.

*2) If* $p = 2$, *then* $\boldsymbol{\delta}^*$ *is given by*

$$\delta_i^* = \begin{cases} P_{[\check{c}_i, \hat{c}_i]}(a_i) & \sum_{i=1}^d (P_{[\check{c}_i, \hat{c}_i]}(a_i))^2 \leq \epsilon^2 \\ P_{[\check{c}_i, \hat{c}_i]}(a_i/(\lambda_2 + 1)) & \text{otherwise}, \end{cases} \tag{15}$$

*where* $\lambda_2 \in (0, \|\mathbf{a}\|_2/\epsilon - 1]$ *is the root of* $\sum_{i=1}^d (P_{[\check{c}_i, \hat{c}_i]}(a_i/(\lambda_2 + 1)))^2 = \epsilon^2$.

*3) If* $p = 0$ *and* $\epsilon \in \mathbb{N}_+$, *then* $\boldsymbol{\delta}^*$ *is given by*

$$\delta_i^* = \begin{cases} \delta_i' & \eta_i \geq [\boldsymbol{\eta}]_\epsilon \\ 0 & \text{otherwise}, \end{cases} \qquad \eta_i = \begin{cases} \sqrt{2a_i \check{c}_i - \check{c}_i^2} & a_i < \check{c}_i \\ \sqrt{2a_i \hat{c}_i - \hat{c}_i^2} & a_i > \hat{c}_i \\ |a_i| & \text{otherwise}. \end{cases} \tag{16}$$

*where* $[\boldsymbol{\eta}]_\epsilon$ *denotes the* $\epsilon$*-th largest element of* $\boldsymbol{\eta}$, *and* $\delta_i' = P_{[\check{c}_i, \hat{c}_i]}(a_i)$.

*Proof of Proposition 1:*

$\ell_1$ **norm**   When we find the Euclidean projection of $\mathbf{a}$ onto the set $\mathcal{X}$, we solve

$$\begin{aligned} \underset{\boldsymbol{\delta}}{\text{minimize}} \quad & \tfrac{1}{2}\|\boldsymbol{\delta} - \mathbf{a}\|_2^2 + I_{[\check{\mathbf{c}}, \hat{\mathbf{c}}]}(\boldsymbol{\delta}) \\ \text{subject to} \quad & \|\boldsymbol{\delta}\|_1 \leq \epsilon, \end{aligned} \tag{17}$$

where $I_{[\check{\mathbf{c}}, \hat{\mathbf{c}}]}(\cdot)$ is the indicator function of the set $[\check{\mathbf{c}}, \hat{\mathbf{c}}]$. The Langragian of this problem is

$$L = \frac{1}{2}\|\boldsymbol{\delta} - \mathbf{a}\|_2^2 + I_{[\check{\mathbf{c}}, \hat{\mathbf{c}}]}(\boldsymbol{\delta}) + \lambda_1(\|\boldsymbol{\delta}\|_1 - \epsilon) \tag{18}$$

$$= \sum_{i=1}^d (\frac{1}{2}(\delta_i - a_i)^2 + \lambda_1|\delta_i| + I_{[\check{c}_i, \hat{c}_i]}(\delta_i)) - \lambda_1 \epsilon. \tag{19}$$

The minimizer $\boldsymbol{\delta}^*$ minimizes the Lagrangian, it is obtained by elementwise soft-thresholding

$$\delta_i^* = P_{[\check{c}_i, \hat{c}_i]}(\text{sign}(a_i) \max\{|a_i| - \lambda_1, 0\}).$$

where $\mathbf{x}_i$ is the $i$th element of a vector $\mathbf{x}$, $P_{[\check{c}_i, \hat{c}_i]}(\cdot)$ is the clip function over the interval $[\check{c}_i, \hat{c}_i]$.

The primal, dual feasibility and complementary slackness are

$$\lambda_1 = 0, \|\boldsymbol{\delta}\|_1 = \sum_{i=1}^d |\delta_i| = \sum_{i=1}^d |P_{[\check{c}_i, \hat{c}_i]}(a_i)| \leq \epsilon \tag{20}$$

$$\textbf{or } \lambda_1 > 0, \|\boldsymbol{\delta}\|_1 = \sum_{i=1}^d |\delta_i| = \sum_{i=1}^d |P_{[\check{c}_i, \hat{c}_i]}(\text{sign}(a_i) \max\{|a_i| - \lambda_1, 0\})| = \epsilon. \tag{21}$$

If $\sum_{i=1}^d |P_{[\check{c}_i, \hat{c}_i]}(a_i)| \leq \epsilon$, $\delta_i^* = P_{[\check{c}_i, \hat{c}_i]}(a_i)$. Otherwise $\delta_i^* = P_{[\check{c}_i, \hat{c}_i]}(\text{sign}(a_i) \max\{|a_i| - \lambda_1, 0\})$, where $\lambda_1$ is given by the root of the equation $\sum_{i=1}^d |P_{[\check{c}_i, \hat{c}_i]}(\text{sign}(a_i) \max\{|a_i| - \lambda_1, 0\})| = \epsilon$. Bisection method can be used to solve the above equation for $\lambda_1$, starting with the initial interval $(0, \max_i |a_i| - \epsilon/d]$. Since $\sum_{i=1}^d |P_{[\check{c}_i, \hat{c}_i]}(\text{sign}(a_i) \max\{|a_i| - 0, 0\})| = \sum_{i=1}^d |P_{[\check{c}_i, \hat{c}_i]}(a_i)| > \epsilon$ in this case, and $\sum_{i=1}^d |P_{[\check{c}_i, \hat{c}_i]}(\text{sign}(a_i) \max\{|a_i| - \max_i |a_i| + \epsilon/d, 0\})| \leq \sum_{i=1}^d |P_{[\check{c}_i, \hat{c}_i]}(\text{sign}(a_i)(\epsilon/d))| \leq \sum_{i=1}^d (\epsilon/d) = \epsilon$.

$\ell_2$ **norm** When we find the Euclidean projection of $\mathbf{a}$ onto the set $\mathcal{X}$, we solve

$$\begin{aligned} \underset{\boldsymbol{\delta}}{\text{minimize}} \quad & \|\boldsymbol{\delta} - \mathbf{a}\|_2^2 + I_{[\check{\mathbf{c}},\hat{\mathbf{c}}]}(\boldsymbol{\delta}) \\ \text{subject to} \quad & \|\boldsymbol{\delta}\|_2^2 \leq \epsilon^2, \end{aligned} \tag{22}$$

where $I_{[\check{\mathbf{c}},\hat{\mathbf{c}}]}(\cdot)$ is the indicator function of the set $[\check{\mathbf{c}},\hat{\mathbf{c}}]$. The Langragian of this problem is

$$L = \|\boldsymbol{\delta} - \mathbf{a}\|_2^2 + I_{[\check{\mathbf{c}},\hat{\mathbf{c}}]}(\boldsymbol{\delta}) + \lambda_2(\|\boldsymbol{\delta}\|_2^2 - \epsilon^2) \tag{23}$$

$$= \sum_{i=1}^d ((\delta_i - a_i)^2 + \lambda_2 \delta_i^2 + I_{[\check{c}_i,\hat{c}_i]}(\delta_i)) - \lambda_2 \epsilon^2. \tag{24}$$

The minimizer $\boldsymbol{\delta}^*$ minimizes the Lagrangian, it is

$$\delta_i^* = P_{[\check{c}_i,\hat{c}_i]}(\frac{1}{\lambda_2 + 1} a_i).$$

The primal, dual feasibility and complementary slackness are

$$\lambda_2 = 0, \|\boldsymbol{\delta}\|_2^2 = \sum_{i=1}^d \delta_i^2 = \sum_{i=1}^d (P_{[\check{c}_i,\hat{c}_i]}(a_i))^2 \leq \epsilon^2 \tag{25}$$

$$\textbf{or } \lambda_2 > 0, \|\boldsymbol{\delta}\|_2^2 = \sum_{i=1}^d \delta_i^2 = (P_{[\check{c}_i,\hat{c}_i]}(\frac{1}{\lambda_2 + 1} a_i))^2 = \epsilon^2. \tag{26}$$

If $\sum_{i=1}^d (P_{[\check{c}_i,\hat{c}_i]}(a_i))^2 \leq \epsilon^2$, $\delta_i^* = P_{[\check{c}_i,\hat{c}_i]}(a_i)$. Otherwise $\delta_i^* = P_{[\check{c}_i,\hat{c}_i]}\left(\frac{1}{\lambda_2+1} a_i\right)$, where $\lambda_2$ is given by the root of the equation $\sum_{i=1}^d (P_{[\check{c}_i,\hat{c}_i]}(\frac{1}{\lambda_2+1} a_i))^2 = \epsilon^2$. Bisection method can be used to solve the above equation for $\lambda_2$, starting with the initial interval $(0, \sqrt{\sum_{i=1}^d (a_i)^2}/\epsilon - 1]$. Since $\sum_{i=1}^d (P_{[\check{c}_i,\hat{c}_i]}(\frac{1}{0+1} a_i))^2 = \sum_{i=1}^d (P_{[\check{c}_i,\hat{c}_i]}(a_i))^2 > \epsilon^2$ in this case, and $\sum_{i=1}^d (P_{[\check{c}_i,\hat{c}_i]}(\frac{1}{\lambda_2+1} a_i))^2 = \sum_{i=1}^d (P_{[\check{c}_i,\hat{c}_i]}(\epsilon a_i/\sqrt{\sum_{i=1}^d (a_i)^2}))^2 \leq \epsilon^2 \sum_{i=1}^d (a_i)^2/(\sqrt{\sum_{i=1}^d (a_i)^2})^2 = \epsilon^2$.

$\ell_0$ **norm** For $\ell_0$ norm in $\mathcal{X}$, it is independent to the box constraint. So we can clip $\mathbf{a}$ to the box constraint first, which is $\delta_i' = P_{[\check{c}_i,\hat{c}_i]}(a_i)$, and then project it onto $\ell_0$ norm.

We find the additional Euclidean distance of every element in $\mathbf{a}$ and zero after they are clipped to the box constraint, which is

$$\eta_i = \begin{cases} \sqrt{a_i^2 - (a_i - \check{c}_i)^2} & a_i < \check{c}_i \\ \sqrt{a_i^2 - (a_i - \hat{c}_i)^2} & a_i > \hat{c}_i \\ |a_i| & \text{otherwise.} \end{cases} \tag{27}$$

It can be equivalently written as

$$\eta_i = \begin{cases} \sqrt{2a_i\check{c}_i - \check{c}_i^2} & a_i < \check{c}_i \\ \sqrt{2a_i\hat{c}_i - \hat{c}_i^2} & a_i > \hat{c}_i \\ |a_i| & \text{otherwise.} \end{cases} \tag{28}$$

To derive the Euclidean projection onto $\ell_0$ norm, we find the $\epsilon$-th largest element in $\boldsymbol{\eta}$ and call it $[\boldsymbol{\eta}]_\epsilon$. We keep the elements whose corresponding $\eta_i$ is above or equals to $\epsilon$-th, and set rest to zeros. The closed-form solution is given by

$$\delta_i^* = \begin{cases} \delta_i' & \eta_i \geq [\boldsymbol{\eta}]_\epsilon \\ 0 & \text{otherwise.} \end{cases} \tag{29}$$

$\square$

**Difference with [25, Proposition 4.1].** We remark that [25] discussed a relevant problem of generating $\ell_p$-norm based adversarial examples under box and linearized classification constraints. The key difference between our proof and that of [25, Proposition 4.1] is summarized below. First, we place $\ell_p$ norm as a hard constraint rather than minimizing it in the objective function. This difference will make our Lagrangian function more involved with a newly introduced non-negative Lagrangian multiplier. Second, the problem of our interest is projection onto the intersection of box and $\ell_p$ constraints. Such a projection step can then be combined with an attack loss (no need of linearization) for generating adversarial examples. Third, we cover the case of $\ell_0$ norm.

## B Proof of Lemma 1

**Lemma 1.** *Problem (11) is equivalent to*

$$\underset{\boldsymbol{\theta}}{\text{minimize}}\ \mathbb{E}_{(\mathbf{x},\mathbf{y})\in\mathcal{D}}\ \underset{\mathbf{w}\in\mathcal{P},\{\boldsymbol{\delta}_i\in\mathcal{X}_i\}}{\text{maximize}}\ \sum_{i=1}^{K} w_i f_{\text{tr}}(\boldsymbol{\theta}, \boldsymbol{\delta}_i; \mathbf{x}, y),$$

*where $\mathbf{w} \in \mathbb{R}^K$ represent domain weights, and $\mathcal{P}$ has been defined in (1).*

***Proof of Lemma 1:***

Similar to (1), problem (11) is equivalent to

$$\underset{\boldsymbol{\theta}}{\text{minimize}}\ \mathbb{E}_{(\mathbf{x},\mathbf{y})\in\mathcal{D}}\ \underset{\mathbf{w}\in\mathcal{P}}{\text{maximize}} \sum_{i=1}^{K} w_i F_i(\boldsymbol{\theta}). \tag{30}$$

Recall that $F_i(\boldsymbol{\theta}) := \text{maximize}_{\boldsymbol{\delta}_i \in \mathcal{X}_i}\ f_{\text{tr}}(\boldsymbol{\theta}, \boldsymbol{\delta}_i; \mathbf{x}, y)$, problem can then be written as

$$\underset{\boldsymbol{\theta}}{\text{minimize}}\ \mathbb{E}_{(\mathbf{x},\mathbf{y})\in\mathcal{D}}\ \underset{\mathbf{w}\in\mathcal{P}}{\text{maximize}} \sum_{i=1}^{K} [w_i \underset{\boldsymbol{\delta}_i \in \mathcal{X}_i}{\text{maximize}}\ f_{\text{tr}}(\boldsymbol{\theta}, \boldsymbol{\delta}_i; \mathbf{x}, y)]. \tag{31}$$

According to proof by contradiction, it is clear that problem (31) is equivalent to

$$\underset{\boldsymbol{\theta}}{\text{minimize}}\ \mathbb{E}_{(\mathbf{x},\mathbf{y})\in\mathcal{D}}\ \underset{\mathbf{w}\in\mathcal{P},\{\boldsymbol{\delta}_i\in\mathcal{X}_i\}}{\text{maximize}} \sum_{i=1}^{K} w_i f_{\text{tr}}(\boldsymbol{\theta}, \boldsymbol{\delta}_i; \mathbf{x}, y). \tag{32}$$

$\square$

# C  Alternating Multi-step PGD (AMPGD) for Generalized AT

In this section, we present the full **a**lternating **m**ulti-step **p**rojected **g**radient **d**escent (AMPGD) algorithm to solve the problem (13), which is repeated as follows

$$\underset{\boldsymbol{\theta}}{\text{minimize}} \ \mathbb{E}_{(\mathbf{x},\mathbf{y})\in\mathcal{D}} \ \underset{\mathbf{w}\in\mathcal{P},\{\boldsymbol{\delta}_i\in\mathcal{X}_i\}}{\text{maximize}} \ \psi(\boldsymbol{\theta},\mathbf{w},\{\boldsymbol{\delta}_i\})$$
$$\psi(\boldsymbol{\theta},\mathbf{w},\{\boldsymbol{\delta}_i\}) := \sum_{i=1}^{K} w_i f_{\text{tr}}(\boldsymbol{\theta},\boldsymbol{\delta}_i;\mathbf{x},y) - \frac{\gamma}{2}\|\mathbf{w}-\mathbf{1}/K\|_2^2$$

---

**Algorithm 3** AMPGD to solve problem (13)

---

1: Input: given $\boldsymbol{\theta}^{(0)}$, $\mathbf{w}^{(0)}$, $\boldsymbol{\delta}^{(0)}$ and $K > 0$.
2: **for** $t = 1, 2, \ldots, T$ **do**
3:     given $\mathbf{w}^{(t-1)}$ and $\boldsymbol{\delta}^{(t-1)}$, perform SGD to update $\boldsymbol{\theta}^{(t)}$
4:     given $\boldsymbol{\theta}^{(t)}$, perform $R$-step PGD to update $\mathbf{w}^{(t)}$ and $\boldsymbol{\delta}^{(t)}$
5: **end for**

---

Problem (13) is in a more general non-convex non-concave min-max setting, where the inner maximization involves both domain weights $\mathbf{w}$ and adversarial perturbations $\{\boldsymbol{\delta}_i\}$. It was shown in [46] that the multi-step PGD is required for inner maximization in order to approximate the near-optimal solution. This is also in the similar spirit of AT [40], which executed multi-step PGD attack during inner maximization. We summarize AMPGD in Algorithm 3. At step 4 of Algorithm 3, each PGD step to update $\mathbf{w}$ and $\boldsymbol{\delta}$ can be decomposed as

$$\mathbf{w}_r^{(t)} = \text{proj}_{\mathcal{P}}\left(\mathbf{w}_{r-1}^{(t)} + \beta\nabla_{\mathbf{w}}\psi(\boldsymbol{\theta}^{(t)},\mathbf{w}_{r-1}^{(t)},\{\boldsymbol{\delta}_{i,r-1}^{(t)}\})\right), \forall r \in [R],$$
$$\boldsymbol{\delta}_{i,r}^{(t)} = \text{proj}_{\mathcal{X}_i}\left(\boldsymbol{\delta}_{i,r-1}^{(t)} + \beta\nabla_{\boldsymbol{\delta}}\psi(\boldsymbol{\theta}^{(t)},\mathbf{w}_{r-1}^{(t)},\{\boldsymbol{\delta}_{i,r-1}^{(t)}\})\right), \forall r,i \in [R],[K]$$

where let $\mathbf{w}_1^{(t)} := \mathbf{w}^{(t-1)}$ and $\boldsymbol{\delta}_{i,1}^{(t)} := \boldsymbol{\delta}_i^{(t-1)}$. Here the superscript $t$ represents the iteration index of AMPGD, and the subscript $r$ denotes the iteration index of $R$-step PGD. Clearly, the above projection operations can be derived for closed-form expressions through (9) and Lemma 1. To the best of our knowledge, it is still an open question to build theoretical convergence guarantees for solving the general non-convex non-concave min-max problem like (13), except the work [46] which proposed $O(1/T)$ convergence rate if the objective function satisfies a strict Polyak-Łojasiewicz condition [29].

# D  Experiment Setup

## D.1  Model Architectures and Training Details

For a comprehensive evaluation of proposed algorithms, we adopt a set of diverse DNN models (Model A to H), including multi-layer perceptrons (MLP), All-CNNs [61], LeNet [30], LeNetV2[3], VGG16 [58], ResNet50 [24], Wide-ResNet [40] and GoogLeNet [63]. For the last four models, we use the exact same architecture as original papers and evaluate them only on CIFAR-10 dataset. The details for model architectures are provided in Table A1. For compatibility with our framework, we implement and train these models based on the strategies adopted in pytorch-cifar[4] and achieve comparable performance on clean images; see Table A2. To foster reproducibility, all the trained models are publicly accessible in the anonymous link. Specifically, we trained MNIST classifiers for 50 epochs with Adam and a constant learning rate of 0.001. For CIFAR-10 classifers, the models are trained for 250 epochs with SGD (using 0.8 nesterov momentum, weight decay $5e^{-4}$). The learning rate is reduced at epoch 100 and 175 with a decay rate of 0.1. The initial learning rate is set as 0.01 for models {A, B, C, D, H} and 0.1 for {E, F, G}. Note that no data augmentation is employed in the training.

Table A1: Neural network architectures used on the MNIST and CIFAR-10 dataset. Conv: convolutional layer, FC: fully connected layer, Globalpool: global average pooling layer.

| **A** (MLP) | **B** (All-CNNs [61]) | **C** (LeNet [30]) | **D** (LeNetV2) |
|---|---|---|---|
| FC(128) + Relu | Conv([32, 64], 3, 3) + Relu | Conv(6, 5, 5) + Relu | Conv(32, 3, 3) + Relu |
| FC(128) + Relu | Conv(128, 3, 3) + Dropout(0.5) | Maxpool(2, 2) | Maxpool(2, 2) |
| FC(64) + Relu | Conv([128, 128], 3, 3) + Relu | Conv(16, 5, 5) + Relu | Conv(64, 3, 3) + Relu |
| FC(10) | Conv(128, 3, 3) + Dropout(0.5) | Maxpool(2, 2) | Maxpool(2, 2) |
| Softmax | Conv(128, 3, 3) + Relu | FC(120) + Relu | FC(128) + Relu |
| | Conv(128, 1, 1) + Relu | FC(84) + Relu | Dropout(0.25) |
| | Conv(10, 1, 1) + Globalpool | FC(10) | FC(10) |
| | Softmax | Softmax | Softmax |
| **E** (VGG16 [58]) | **F** (ResNet50 [24]) | **G** (Wide-ResNet [40]) | **H** (GoogLeNet [63]) |

Table A2: Clean test accuracy of DNN models on MNIST and CIFAR-10. We roughly derive the model robustness by attacking models separately using FGSM [23]. The adversarial examples are generated by FGSM $\ell_\infty$-attack ($\epsilon = 0.2$).

| MNIST | | | CIFAR-10 | | | | | |
|---|---|---|---|---|---|---|---|---|
| Model | Acc. | FGSM | Model | Acc. | FGSM | Model | Acc. | FGSM |
| A: MLP | 98.20% | 18.92% | A: MLP | 55.36% | 11.25% | E: VGG16 | 87.57% | 10.83% |
| B: All-CNNs | 99.49% | 50.95% | B: All-CNNs | 84.18% | 9.89% | F: ResNet50 | 88.11% | 10.73% |
| C: LeNet | 99.25% | 63.23% | C: LeNet | 64.95% | 14.45% | G: Wide-ResNet | 91.67% | 15.78% |
| D: LeNetV2 | 99.33% | 56.36% | D: LeNetV2 | 74.89% | 9.77% | H: GoogLeNet | 90.92% | 9.91% |

## D.2  Crafting Adversarial Examples

We adopt variant C&W loss in APGDA/PGD as suggested in [40, 13] with a confidence parameter $\kappa = 50$. Cross-entropy loss is also supported in our implementation. The adversarial examples are generated by 20-step PGD/APGDA unless otherwise stated (e.g., 50 steps for ensemble attacks). Note that proposed algorithms are robust and will not be affected largely by the choices of hyperparameters $(\alpha, \beta, \gamma)$. In consequence, we do not finely tune the parameters on the validation set. Specifically, The learning rates $\alpha, \beta$ and regularization factor $\gamma$ for Table 1 are set as - $\ell_0 : \alpha = 1, \beta = \frac{1}{100}, \gamma = 7$, $\ell_1 : \alpha = \frac{1}{4}, \beta = \frac{1}{100}, \gamma = 5$, $\ell_2 : \alpha = \frac{1}{10}, \beta = \frac{1}{100}, \gamma = 3$; $\ell_\infty : \alpha = \frac{1}{4}, \beta = \frac{1}{50}, \gamma = 3$. For Table 3, the hyper-parameters are set as $\ell_0 : \alpha = 1, \beta = \frac{1}{150}, \gamma = 1$, $\ell_1 : \alpha = \frac{1}{4}, \beta = \frac{1}{100}, \gamma = 5$, $\ell_2 : \alpha = \frac{1}{8}, \beta = \frac{1}{100}, \gamma = 3$; $\ell_\infty : \alpha = \frac{1}{5}, \beta = \frac{1}{50}, \gamma = 6$.

---

[3]An enhanced version of original LeNet with more layers and units (see Table A1 Model D).
[4]https://github.com/kuangliu/pytorch-cifar

Due to varying model robustness on different datasets, the perturbation magnitudes $\epsilon$ are set separately [11]. For universal perturbation experiments, the $\epsilon$ are set as 0.2 (A, B), 0.3 (C) and 0.25 (D) on MNIST; 0.02 (B, H), 0.35 (E) and 0.05 (D) on CIFAR-10. For generalized AT, the models on MNIST are trained following the same rules in last section, except that training epochs are prolonged to 350 and adversarial examples are crafted for assisting the training with a ratio of 0.5. Our experiment setup is based on CleverHans package[5] and Carlini and Wagner's framework[6].

### D.3 Details of Conducted Data Transformations

To demonstrate the effectiveness of APGDA in generating robust adversarial examples against multiple transformations, we adopt a series of common transformations, including a&b) flipping images horizontally (*flh*) and vertically (*flv*); c) adjusting image brightness (*bri*); d) performing gamma correction (*gam*), e) cropping and re-sizing images (*crop*); f) rotating images (*rot*).

Moreover, both deterministic and stochastic transformations are considered in our experiments. In particular, Table 7 and Table A5 are deterministic settings - *rot*: rotating images 30 degree clockwise; *crop*: cropping images in the center ($0.8 \times 0.8$) and resizing them to $32 \times 32$; *bri*: adjusting the brightness of images with a scale of 0.1; *gam*: performing gamma correction with a value of 1.3. Differently, in Table A4, we introduce randomness for drawing samples from the distribution - *rot*: rotating images randomly from -10 to 10 degree; *crop*: cropping images in the center randomly (from 0.6 to 1.0); other transformations are done with a probability of 0.8. In experiments, we adopt `tf.image` API [7] for processing the images.

---

[5] https://github.com/tensorflow/cleverhans
[6] https://github.com/carlini/nn_robust_attacks
[7] https://www.tensorflow.org/api_docs/python/tf/image

# E   Additional Experiment Results - Robust adversarial attacks

## E.1   Ensemble Attack over Multiple Models

Table 3 and A3 shows the performance of average (ensemble PGD [34]) and min-max (APGDA) strategies for attacking model ensembles. Our min-max approach results in 19.27% and 15.69% averaged improvement on $\text{ASR}_{all}$ over models {A, B, C, D} and {A, E, F, H} on CIFAR-10.

Table A3: Comparison of average and min-max (APGDA) ensemble attack over models {A, E, F, H} on CIFAR-10. Acc (%) represents the test accuracy of classifiers on adversarial examples. The learning rates $\alpha, \beta$ and regularization factor $\gamma$ are set as - $\ell_0 : \alpha = 1, \beta = \frac{1}{150}, \gamma = 1$, $\ell_1 : \alpha = \frac{1}{4}, \beta = \frac{1}{100}, \gamma = 5, \ell_2 : \alpha = \frac{1}{8}, \beta = \frac{1}{100}, \gamma = 3; \ell_\infty : \alpha = \frac{1}{5}, \beta = \frac{1}{50}, \gamma = 6$. The attack iteration for APGDA is set as 50.

| Box constraint | Opt. | $\text{Acc}_A$ | $\text{Acc}_E$ | $\text{Acc}_F$ | $\text{Acc}_H$ | $\text{ASR}_{all}$ | Lift ($\uparrow$) |
|---|---|---|---|---|---|---|---|
| $\ell_0$ ($\epsilon = 70$) | $avg.$ | 27.38 | 6.33 | 7.18 | 6.99 | 66.56 | - |
| | min max | 19.38 | 8.72 | 9.48 | 8.94 | **73.83** | **10.92%** |
| $\ell_1$ ($\epsilon = 30$) | $avg.$ | 30.90 | 2.06 | 1.85 | 1.84 | 66.23 | - |
| | min max | 12.56 | 3.21 | 2.70 | 2.72 | **83.13** | **25.52%** |
| $\ell_2$ ($\epsilon = 1.5$) | $avg.$ | 20.87 | 1.75 | 1.21 | 1.54 | 76.41 | - |
| | min max | 10.26 | 3.15 | 2.24 | 2.37 | **84.99** | **11.23%** |
| $\ell_\infty$ ($\epsilon = 0.03$) | $avg.$ | 25.75 | 2.59 | 1.66 | 2.27 | 70.54 | - |
| | min max | 13.47 | 3.79 | 3.15 | 3.48 | **81.17** | **15.07%** |

To perform a boarder evaluation, we repeat the above experiments ($\ell_\infty$ norm) under different $\epsilon$ in Figure A1. The ASR of min-max strategy is consistently better or on part with the average strategy. Moreover, APGDA achieves more significant improvement when moderate $\epsilon$ is chosen: MNIST ($\epsilon \in [0.15, 0.25]$) and CIFAR-10 ($\epsilon \in [0.03, 0.05]$).

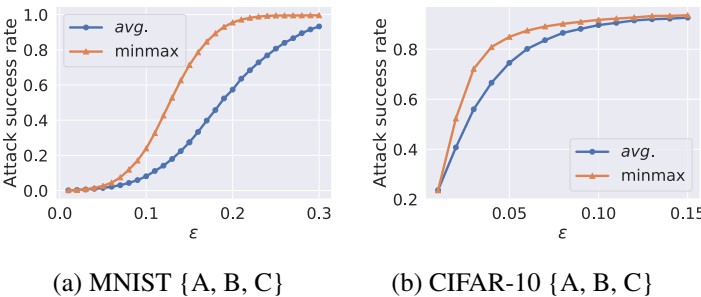

(a) MNIST {A, B, C}          (b) CIFAR-10 {A, B, C}

Figure A1: ASR of average and min-max $\ell_\infty$ ensemble attack versus maximum perturbation magnitude $\epsilon$.

## E.2   Robust Adversarial Attack over Data Transformations

Table A4 compare the performance of average (EOT [5]) and min-max (APGDA) strategies. Our approach results in 4.31% averaged lift over four models {A, B, C, D} on CIFAR-10 under given stochastic and deterministic transformation sets.

## E.3   Analysis of Regularization on Probability Simplex

To further explore the utility of quadratic regularizer on the probability simplex in proposed min-max framework, we conducted sensitivity analysis on $\gamma$ and show how the proposed regularization affects the eventual performance (Figure A2a) taking ensemble attack as an example. The experimental setting is the same as Table 1 except for altering the value of $\gamma$ from 0 to 10. Figure A2a shows that too small or too large $\gamma$ leads to relative weak performance due to the unstable convergence and penalizing too much for average case. When $\gamma$ is around 4, APGDA will achieve the best performance so we adopted this value in the experiments (Table 1). Moreover, when $\gamma \rightarrow \infty$, the regularizer term dominates the optimization objective and it becomes the average case.

Table A4: Comparison of average and min-max optimization on robust attack over multiple data transformations on CIFAR-10. Note that all data transformations are conducted stochastically with a probability of 0.8, except for *crop* which randomly crops a central area from original image and re-size it into $32 \times 32$. The adversarial examples are generated by 20-step $\ell_\infty$-APGDA ($\epsilon = 0.03$) with $\alpha = \frac{1}{2}, \beta = \frac{1}{100}$ and $\gamma = 10$.

| Model | Opt. | $\text{Acc}_{ori}$ | $\text{Acc}_{flh}$ | $\text{Acc}_{flv}$ | $\text{Acc}_{bri}$ | $\text{Acc}_{crop}$ | $\text{ASR}_{avg}$ | $\text{ASR}_{gp}$ | Lift ($\uparrow$) |
|---|---|---|---|---|---|---|---|---|---|
| A | *avg.* | 11.55 | 21.60 | 13.64 | 12.30 | 22.37 | 83.71 | 55.97 | - |
|   | min max | 13.06 | 18.90 | 13.43 | 13.90 | 20.27 | 84.09 | **59.17** | **5.72%** |
| B | *avg.* | 6.74 | 11.55 | 10.33 | 6.59 | 18.21 | 89.32 | 69.52 | - |
|   | min max | 8.19 | 11.13 | 10.31 | 8.31 | 16.29 | 89.15 | **71.18** | **2.39%** |
| C | *avg.* | 8.23 | 17.47 | 13.93 | 8.54 | 18.83 | 86.60 | 58.85 | - |
|   | min max | 9.68 | 13.45 | 13.41 | 9.95 | 18.23 | 87.06 | **61.63** | **4.72%** |
| D | *avg.* | 8.67 | 19.75 | 11.60 | 8.46 | 19.35 | 86.43 | 60.96 | - |
|   | min max | 10.43 | 16.41 | 12.14 | 10.15 | 17.64 | 86.65 | **63.64** | **4.40%** |

Table A5: Comparison of average and min-max optimization on robust attack over multiple data transformations on CIFAR-10. Here a new rotation (*rot*) transformation is introduced, where images are rotated 30 degrees clockwise. Note that all data transformations are conducted with a probability of 1.0. The adversarial examples are generated by 20-step $\ell_\infty$-APGDA ($\epsilon = 0.03$) with $\alpha = \frac{1}{2}, \beta = \frac{1}{100}$ and $\gamma = 10$.

| Model | Opt. | $\text{Acc}_{ori}$ | $\text{Acc}_{flh}$ | $\text{Acc}_{flv}$ | $\text{Acc}_{bri}$ | $\text{Acc}_{gam}$ | $\text{Acc}_{crop}$ | $\text{Acc}_{rot}$ | $\text{ASR}_{avg}$ | $\text{ASR}_{gp}$ | Lift ($\uparrow$) |
|---|---|---|---|---|---|---|---|---|---|---|---|
| A | *avg.* | 11.06 | 22.37 | 14.81 | 12.32 | 10.92 | 20.40 | 15.89 | 84.60 | 49.24 | - |
|   | min max | 13.51 | 18.84 | 14.03 | 15.20 | 13.00 | 18.03 | 14.79 | 84.66 | **52.31** | **6.23%** |
| B | *avg.* | 5.55 | 11.96 | 9.97 | 5.63 | 5.94 | 16.42 | 11.47 | 90.44 | 65.18 | - |
|   | min max | 6.75 | 9.13 | 10.56 | 6.72 | 7.11 | 12.23 | 10.80 | 90.96 | **70.38** | **7.98%** |
| C | *avg.* | 7.65 | 22.30 | 15.82 | 8.17 | 8.07 | 15.44 | 15.09 | 86.78 | 49.67 | - |
|   | min max | 9.05 | 15.10 | 14.57 | 9.57 | 9.31 | 14.11 | 14.23 | 87.72 | **55.37** | **11.48%** |
| D | *avg.* | 8.22 | 20.88 | 13.49 | 7.91 | 8.71 | 16.33 | 14.98 | 87.07 | 53.52 | - |
|   | min max | 10.17 | 14.65 | 13.62 | 10.03 | 10.35 | 14.36 | 13.82 | 87.57 | **57.36** | **7.17%** |

# F   Additional Experiment Results - Adversarial training against multiple types of adversarial attacks

**Adversarial Training Details:**   Following the state-of-the-art approach MSD [41], we present experimental results of generalized AT to achieve simultaneous robustness to $\ell_\infty$, $\ell_2$, and $\ell_1$ perturbations on the MNIST and CIFAR-10 datasets. Specifically, we adopted the same architectures as [41] four layer convolutional networks on MNIST and the pre-activation version of the ResNet18 [24]. The perturbation radius $\epsilon$ for $(\ell_\infty, \ell_2, \ell_1)$ balls is set as $(0.3, 2.0, 10)$ and $(0.03, 0.5, 12)$ on MNIST and CIFAR-10 following [41]. For MNIST models, all models are trained 15 epochs with the Adam optimizer. We used a variation of the learning rate schedule from [60] - piecewise linear schedule from 0 to $10^{-3}$ over the first 6 epochs, and down to 0 over the last 9 epochs. For CIFAR-10 models, we

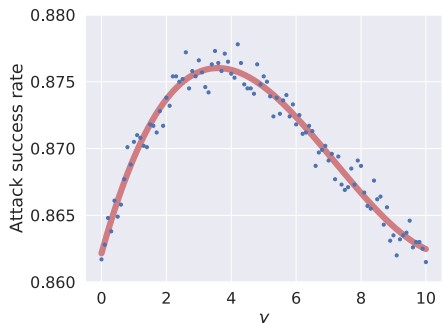

Figure A2: Sensitivity analysis of the regularizer $\frac{\gamma}{2}\|\mathbf{w} - 1/K\|_2^2$ on the probability simplex. The experimental setting is the same as Table 1 except for altering the value of $\gamma$.

trained all the models for 50 epochs and used the SGD optimizer with momentum 0.9 and weight decay $5 \times 10^{-4}$. The learning rate schedule rate is piecewise linear from 0 to 0.1 over the first 20 epochs, down to 0.005 over the next 20 epochs, and finally back down to 0 in the last 10 epochs.

**Evaluation Setup:** To make fair comparisons with MSD [41], we implemented AMPGD based on the public codebase[8] and followed the exact evaluation settings. Specifically, for $\ell_\infty$ attacks, we use FGSM [22], PGD attack [40] and Momentum Iterative Method [19]. For $\ell_2$ attacks, we use PGD attack, the Gaussian noise attack [52], the boundary attack [9] (Brendel et al., 2017), DeepFool [45], the pointwise attack [55], DDN-based attack [54] and C&W attack [13]. For $\ell_1$ attacks, we use the $\ell_1$ PGD attack, the salt & pepper attack [52] and the pointwise attack [55]. Moreover, we also incorporate the state-of-the-art AutoAttack [18] for a more comprehensive evaluation under mixed $\ell_p$ perturbations.

**Experimental Results:** The complete adversarial accuracy results on $\ell_p$ attacks and the union of them are shown in Table A6. As we can see, our AMPGD approach leads to a consistent and significant improvement on MNIST. Compared to MSD, we found that our AMPGD emphasize more on defending the strongest adversary - $\ell_\infty$ PGD thus avoiding biased by one particular perturbation model. This observation is also consistent to the learning curves in Figure 3.

Table A6: Summary of adversarial robustness on MNIST.

|  | $L_\infty$-AT | $L_2$-AT | $L_1$-AT | MAX [65] | AVG [65] | MSD [41] | AMPGD |
|---|---|---|---|---|---|---|---|
| Clean Accuracy | 99.1% | 99.2% | 99.3% | 98.6% | 99.1% | 98.3% | 98.3% |
| $\ell_\infty$ Attacks ($\epsilon = 0.3$) [41] | 90.3% | 0.4% | 0.0% | 51.0% | 65.2% | 62.7% | 76.1% |
| $\ell_2$ Attacks ($\epsilon = 2.0$) [41] | 13.6% | 69.2% | 38.5% | 61.9% | 60.1% | 67.9% | 70.2% |
| $\ell_1$ Attacks ($\epsilon = 10$) [41] | 4.2% | 43.4% | 70.0% | 52.6% | 39.2% | 65.0% | 67.2% |
| All Attacks [41] | 3.7% | 0.4% | 0.0% | 42.1% | 34.9% | 58.4% | **64.1%** |
| AA ($\ell_\infty, \epsilon = 0.3$) [18] | 89.5% | 0.0% | 0.0% | 55.0% | 52.8% | 56.6% | 74.4% |
| AA ($\ell_2, \epsilon = 2.0$) [18] | 3.5% | 67.6% | 37.3% | 56.9% | 55.8% | 68.1% | 63.8% |
| AA ($\ell_1, \epsilon = 10$) [18] | 2.4% | 60.1% | 71.9% | 46.5% | 40.7% | 70.0% | 60.5% |
| AA (all attacks) [18] | 1.7% | 0.0% | 0.0% | 36.9% | 30.5% | 55.9% | **59.3%** |
| AA+ ($\ell_\infty, \epsilon = 0.3$) [18] | 89.6% | 0.0% | 0.0% | 54.4% | 52.4% | 55.7% | 74.3% |
| AA+ ($\ell_2, \epsilon = 2.0$) [18] | 2.1% | 67.4% | 36.8% | 55.9% | 53.8% | 67.3% | 61.9% |
| AA+ ($\ell_1, \epsilon = 10$) [18] | 1.8% | 60.4% | 71.4% | 42.3% | 36.7% | 68.6% | 59.8% |
| AA+ (all attacks) [18] | 1.2% | 0.0% | 0.0% | 34.3% | 28.8% | 54.8% | **58.3%** |

---

[8]https://github.com/locuslab/robust_union

# G  Interpreting "Image Robustness" with Domain Weights w

Tracking *domain weight* $w$ of the probability simplex from our algorithms is an exclusive feature of solving problem 1. In Sec. 4, we show the strength of $w$ in understanding the procedure of optimization and interpreting the adversarial robustness. Here we would like to show the usage of $w$ in measuring "image robustness" on devising universal perturbation to multiple input samples. Table A7 and A8 show the image groups on MNIST with weight $w$ in APGDA and two metrics (distortion of $\ell 2$-C&W, minimum $\epsilon$ for $\ell_\infty$-PGD) of measuring the difficulty of attacking single images. The binary search is utilized to searching for the minimum perturbation.

Although adversaries need to consider a trade-off between multiple images while devising universal perturbation, we find that weighting factor $w$ in APGDA is highly correlated under different $\ell_p$ norms. Furthermore, $w$ is also highly related to minimum distortion required for attacking a single image successfully. It means the inherent "image robustness" exists and effects the behavior of generating universal perturbation. Larger weight $w$ usually indicates an image with higher robustness (e.g., fifth 'zero' in the first row of Table A7), which usually corresponds to the MNIST letter with clear appearance (e.g., bold letter).

Table A7: Interpretability of domain weight $w$ for universal perturbation to multiple inputs on MNIST (*Digit 0 to 4*). Domain weight $w$ for different images under $\ell_p$-norm ($p = 0, 1, 2, \infty$) and two metrics measuring the difficulty of attacking single image are recorded, where dist. ($\ell_2$) denotes the the minimum distortion of successfully attacking images using C&W ($\ell_2$) attack; $\epsilon_{\min}$ ($\ell_\infty$) denotes the minimum perturbation magnitude for $\ell_\infty$-PGD attack.

| | Image |  | | | | | | | | | |
|---|---|---|---|---|---|---|---|---|---|---|---|
| Weight | $\ell_0$ | 0. | 0. | 0. | 0. | 1.000 | 0.248 | 0.655 | 0.097 | 0. | 0. |
| | $\ell_1$ | 0. | 0. | 0. | 0. | 1.000 | 0.07 | 0.922 | 0. | 0. | 0. |
| | $\ell_2$ | 0. | 0. | 0. | 0. | 1.000 | 0.441 | 0.248 | 0.156 | 0.155 | 0. |
| | $\ell_\infty$ | 0. | 0. | 0. | 0. | 1.000 | 0.479 | 0.208 | 0.145 | 0.168 | 0. |
| Metric | dist.(C&W $\ell_2$) | 1.839 | 1.954 | 1.347 | 1.698 | 3.041 | 1.545 | 1.982 | 2.178 | 2.349 | 1.050 |
| | $\epsilon_{\min}$ ($\ell_\infty$) | 0.113 | 0.167 | 0.073 | 0.121 | 0.199 | 0.167 | 0.157 | 0.113 | 0.114 | 0.093 |

| | Image |  | | | | | | | | | |
|---|---|---|---|---|---|---|---|---|---|---|---|
| Weight | $\ell_0$ | 0. | 0. | 0.613 | 0.180 | 0.206 | 0. | 0. | 0.223 | 0.440 | 0.337 |
| | $\ell_1$ | 0. | 0. | 0.298 | 0.376 | 0.327 | 0. | 0. | 0.397 | 0.433 | 0.169 |
| | $\ell_2$ | 0. | 0. | 0.387 | 0.367 | 0.246 | 0. | 0.242 | 0.310 | 0.195 | 0.253 |
| | $\ell_\infty$ | 0.087 | 0.142 | 0.277 | 0.247 | 0.246 | 0. | 0.342 | 0.001 | 0.144 | 0.514 |
| Metric | dist.(C&W $\ell_2$) | 1.090 | 1.182 | 1.327 | 1.458 | 0.943 | 0.113 | 1.113 | 1.357 | 1.474 | 1.197 |
| | $\epsilon_{\min}$ ($\ell_\infty$) | 0.075 | 0.068 | 0.091 | 0.105 | 0.096 | 0.015 | 0.090 | 0.076 | 0.095 | 0.106 |

| | Image |  | | | | | | | | | |
|---|---|---|---|---|---|---|---|---|---|---|---|
| Weight | $\ell_0$ | 0. | 1.000 | 0. | 0. | 0. | 0. | 0. | 0.909 | 0. | 0.091 |
| | $\ell_1$ | 0. | 1.000 | 0. | 0. | 0. | 0. | 0. | 0.843 | 0. | 0.157 |
| | $\ell_2$ | 0. | 0.892 | 0. | 0. | 0.108 | 0. | 0. | 0.788 | 0. | 0.112 |
| | $\ell_\infty$ | 0. | 0.938 | 0. | 0. | 0.062 | 0. | 0. | 0.850 | 0. | 0.150 |
| Metric | dist.(C&W $\ell_2$) | 1.335 | 2.552 | 2.282 | 1.229 | 1.884 | 1.928 | 1.439 | 2.312 | 1.521 | 2.356 |
| | $\epsilon_{\min}$ ($\ell_\infty$) | 0.050 | 0.165 | 0.110 | 0.083 | 0.162 | 0.082 | 0.106 | 0.176 | 0.072 | 0.171 |

| | Image |  | | | | | | | | | |
|---|---|---|---|---|---|---|---|---|---|---|---|
| Weight | $\ell_0$ | 0.481 | 0. | 0.378 | 0. | 0. | 0. | 0.352 | 0. | 0. | 0.648 |
| | $\ell_1$ | 0.690 | 0. | 0.310 | 0. | 0. | 0. | 0.093 | 0.205 | 0. | 0.701 |
| | $\ell_2$ | 0.589 | 0.069 | 0.208 | 0. | 0.134 | 0.064 | 0.260 | 0.077 | 0. | 0.600 |
| | $\ell_\infty$ | 0.864 | 0. | 0.084 | 0. | 0.052 | 0.079 | 0.251 | 0.156 | 0. | 0.514 |
| Metric | dist.(C&W $\ell_2$) | 2.267 | 1.656 | 2.053 | 1.359 | 0.861 | 1.733 | 1.967 | 1.741 | 1.031 | 2.413 |
| | $\epsilon_{\min}$ ($\ell_\infty$) | 0.171 | 0.088 | 0.143 | 0.117 | 0.086 | 0.100 | 0.097 | 0.096 | 0.038 | 0.132 |

| | Image |  | | | | | | | | | |
|---|---|---|---|---|---|---|---|---|---|---|---|
| Weight | $\ell_0$ | 0. | 0. | 0.753 | 0. | 0.247 | 0. | 0. | 0. | 1.000 | 0. |
| | $\ell_1$ | 0.018 | 0. | 0.567 | 0. | 0.416 | 0.347 | 0. | 0. | 0.589 | 0.063 |
| | $\ell_2$ | 0. | 0. | 0.595 | 0. | 0.405 | 0.346 | 0. | 0. | 0.654 | 0. |
| | $\ell_\infty$ | 0. | 0. | 0.651 | 0. | 0.349 | 0.239 | 0. | 0. | 0.761 | 0. |
| Metric | dist.(C&W $\ell_2$) | 1.558 | 1.229 | 1.939 | 0.297 | 1.303 | 0.940 | 1.836 | 1.384 | 1.079 | 2.027 |
| | $\epsilon_{\min}$ ($\ell_\infty$) | 0.084 | 0.088 | 0.122 | 0.060 | 0.094 | 0.115 | 0.103 | 0.047 | 0.125 | 0.100 |

Table A8: Interpretability of domain weight $w$ for universal perturbation to multiple inputs on MNIST (*Digit 5 to 9*). Domain weight $w$ for different images under $\ell_p$-norm ($p = 0, 1, 2, \infty$) and two metrics measuring the difficulty of attacking single image are recorded, where dist. ($\ell_2$) denotes the the minimum distortion of successfully attacking images using C&W ($\ell_2$) attack; $\epsilon_{\min}$ ($\ell_\infty$) denotes the minimum perturbation magnitude for $\ell_\infty$-PGD attack.

| Image |  | | | | | | | | | |
|---|---|---|---|---|---|---|---|---|---|---|
| **Weight** $\ell_0$ | 0. | 0.062 | 0.254 | 0. | 0.684 | 0.457 | 0. | 0. | 0.542 | 0. |
| $\ell_1$ | 0.131 | 0.250 | 0. | 0. | 0.619 | 0.033 | 0.157 | 0.005 | 0.647 | 0.158 |
| $\ell_2$ | 0.012 | 0.164 | 0.121 | 0. | 0.703 | 0.161 | 0.194 | 0. | 0.508 | 0.136 |
| $\ell_\infty$ | 0.158 | 0.008 | 0.258 | 0. | 0.576 | 0.229 | 0.179 | 0. | 0.401 | 0.191 |
| **Metric** dist. ($\ell_2$) | 1.024 | 1.532 | 1.511 | 1.351 | 1.584 | 1.319 | 1.908 | 1.020 | 1.402 | 1.372 |
| $\epsilon_{\min}$ ($\ell_\infty$) | 0.090 | 0.106 | 0.085 | 0.069 | 0.144 | 0.106 | 0.099 | 0.0748 | 0.131 | 0.071 |
| Image |  | | | | | | | | | |
| **Weight** $\ell_0$ | 0.215 | 0. | 0. | 0.194 | 0.590 | 0.805 | 0. | 0. | 0.195 | 0. |
| $\ell_1$ | 0.013 | 0. | 0. | 0.441 | 0.546 | 0.775 | 0. | 0. | 0.225 | 0. |
| $\ell_2$ | 0.031 | 0. | 0. | 0.410 | 0.560 | 0.767 | 0. | 0. | 0.233 | 0. |
| $\ell_\infty$ | 0. | 0. | 0. | 0.459 | 0.541 | 0.854 | 0. | 0. | 0.146 | 0. |
| **Metric** dist. ($\ell_2$) | 1.199 | 0.653 | 1.654 | 1.156 | 1.612 | 2.158 | 0. | 1.063 | 1.545 | 0.147 |
| $\epsilon_{\min}$ ($\ell_\infty$) | 0.090 | 0.017 | 0.053 | 0.112 | 0.158 | 0.159 | 0.020 | 0.069 | 0.145 | 0.134 |
| Image |  | | | | | | | | | |
| **Weight** $\ell_0$ | 0.489 | 0. | 0. | 0.212 | 0.298 | 0.007 | 0.258 | 0.117 | 0.482 | 0.136 |
| $\ell_1$ | 0.525 | 0.190 | 0. | 0.215 | 0.070 | 0.470 | 0.050 | 0.100 | 0.343 | 0.038 |
| $\ell_2$ | 0.488 | 0.165 | 0. | 0.175 | 0.172 | 0.200 | 0.175 | 0.233 | 0.378 | 0.014 |
| $\ell_\infty$ | 0.178 | 0.263 | 0. | 0.354 | 0.205 | 0.258 | 0.207 | 0.109 | 0.426 | 0. |
| **Metric** dist. ($\ell_2$) | 1.508 | 1.731 | 1.291 | 1.874 | 1.536 | 1.719 | 2.038 | 1.417 | 2.169 | 0.848 |
| $\epsilon_{\min}$ ($\ell_\infty$) | 0.110 | 0.125 | 0.089 | 0.126 | 0.095 | 0.087 | 0.097 | 0.084 | 0.135 | 0.077 |
| Image |  | | | | | | | | | |
| **Weight** $\ell_0$ | 0. | 0. | 1.000 | 0. | 0. | 0.246 | 0. | 0. | 0. | 0.754 |
| $\ell_1$ | 0. | 0.180 | 0.442 | 0.378 | 0. | 0.171 | 0. | 0. | 0. | 0.829 |
| $\ell_2$ | 0. | 0.298 | 0.593 | 0.109 | 0. | 0.330 | 0. | 0. | 0. | 0.670 |
| $\ell_\infty$ | 0. | 0.377 | 0.595 | 0.028 | 0. | 0.407 | 0. | 0. | 0. | 0.593 |
| **Metric** dist. ($\ell_2$) | 1.626 | 1.497 | 1.501 | 1.824 | 0.728 | 1.928 | 1.014 | 1.500 | 1.991 | 1.400 |
| $\epsilon_{\min}$ ($\ell_\infty$) | 0.070 | 0.153 | 0.156 | 0.156 | 0.055 | 0.171 | 0.035 | 0.090 | 0.170 | 0.161 |
| Image |  | | | | | | | | | |
| **Weight** $\ell_0$ | 1. | 0. | 0. | 0. | 0. | 0. | 0.665 | 0.331 | 0. | 0.004 |
| $\ell_1$ | 0.918 | 0. | 0.012 | 0. | 0.070 | 0. | 0.510 | 0.490 | 0. | 0. |
| $\ell_2$ | 0.911 | 0. | 0.089 | 0. | 0. | 0. | 0.510 | 0.490 | 0. | 0. |
| $\ell_\infty$ | 0.935 | 0. | 0.065 | 0. | 0. | 0. | 0.665 | 0.331 | 0. | 0.004 |
| **Metric** dist. ($\ell_2$) | 1.961 | 1.113 | 1.132 | 1.802 | 0.939 | 1.132 | 1.508 | 1.335 | 1.033 | 1.110 |
| $\epsilon_{\min}$ ($\ell_\infty$) | 0.144 | 0.108 | 0.083 | 0.103 | 0.079 | 0.041 | 0.090 | 0.103 | 0.083 | 0.044 |