# OpenReview forum: "Adversarial Attack Generation Empowered by Min-Max Optimization"
_NeurIPS.cc/2021/Conference — NeurIPS 2021 Poster_

### Official Review · Reviewer_6whq · 2021-07-13

**Rating:** 5
**Confidence:** 3

**Summary:**

The work proposes a min-max framework APGDA that can take in different models related to adversarial robustness. APGDA can generate model ensemble attack, universal attack over multiple images, and robust attack over data transformations. They perform experiments to evaluate their method.

**Limitations And Societal Impact:**

1. The authors perform many experiments on MNIST, but MNIST is known to be a poor dataset to study adversarial examples, especially in this experiment-based article.

2. As the authors have trained models on MNIST and CIFAR-10, why not demonstrate Table 2 on CIFAR-10? According to Table 1 and Appendix E, their method performs better on MNIST, I suspect it will perform worse on large datasets(e.g., ImageNet).

**Main Review:**

**Originality:**

**Are the tasks or methods new? Is the work a novel combination of well-known techniques?**

It doesn't differ much from previous work (Universal Adversarial Training, Shafahi et al., 2018).

**Is it clear how this work differs from previous contributions? Is related work adequately cited?**

Yes.


**Quality:**

**Is the submission technically sound? Are claims well supported (e.g., by theoretical analysis or experimental results)? Are the methods used appropriate?**

Partly, claims are mainly supported by experiments.

**Is this a complete piece of work or work in progress? Are the authors careful and honest about evaluating both the strengths and weaknesses of their work?**

Yes.


**Clarity:**

**Is the submission clearly written? Is it well organized? (If not, please make constructive suggestions for improving its clarity.)**

Mostly clear.

**Does it adequately inform the reader? (Note that a superbly written paper provides enough information for an expert reader to reproduce its results.)**

Yes.


**Significance:**

**Are the results important? Are others (researchers or practitioners) likely to use the ideas or build on them? Does the submission address a difficult task in a better way than previous work?**

Not sure.

**Does it advance the state of the art in a demonstrable way? Does it provide unique data, unique conclusions about existing data, or a unique theoretical or experimental approach?**

N/A

**Time Spent Reviewing:**

3 hours

---

> ### Author Response · Authors · 2021-08-10
> **Response to Reviewer 6whq**
>
> Thanks for your thoughtful reviews and comments. We are addressing the concerns as follows.
>
> **Q1**: It doesn't differ much from previous work (Universal Adversarial Training, Shafahi et al., 2018). \
> **A1**: We **respectfully disagree**. Our work differs largely with Shafahi et al., 2018 - solving different problems with different methodologies!
> First of all, the main contribution of our work is to introduce the min-max principle in attack design. However, Shafahi et al. proposed to use standard adversarial training (AT) to make the model robust to universal perturbations (where the universal perturbation is generated by **average strategy rather than the min-max strategy that we proposed**). Even on the defense side, we highlight two main differences: (1) Shafahi et al. proposed a min-max based AT by leveraging universal perturbations (rather than per-image perturbation). On the contrary, we generalize AT subject to mixed types of adversarial perturbations (multiple $\ell_p$ perturbations). (2) Our min-max formulation (11) stems from the min-max-max formulation (18), where the last max step is conducted on the **importance weights $\mathbf w$**, which is different from standard AT used by Shafahi et al.
>
> **Q2**: The authors perform many experiments on MNIST, but MNIST is known to be a poor dataset to study adversarial examples, especially in this experiment-based article. \
> **A2**: First, we also conducted many experiments on CIFAR-10 in each setting (three attack tasks and generated adversarial training in defense). Compared to MNIST, we tested more advanced models (e.g., VGG16, Wide-ResNet, GoogLeNet) on CIFAR-10.
>
> Second, MNIST is a good dataset for us to demonstrate the usefulness of our proposed min-max attack design. For example, we use MNIST to visualize the effect of self-adjusted weights on the attack performance (Figure 1 and Table 4). It is clear to see that the larger domain weights correspond to the MNIST letters with clearer appearance, implying that the learnable weights could offer visual interpretability of “image robustness”.
>
> Third, we hesitate to call MNIST a poor dataset for studying adversarial examples. To the best of our knowledge, many seminal works [R1, R2, R3] considered MNIST as a standard dataset. A possible relevant point in [R3] is that one should consider different perturbation radiuses for different datasets; for example, ($\epsilon = 0.2$, $\ell_\infty$ attack) used for MNIST becomes too large for CIFAR-10. In our work, we follow the commonly-used setting of the perturbation radius, 0.2 for MNIST, 0.05 or 0.03 for CIFAR-10.
>
> [R1] Towards Deep Learning Models Resistant to Adversarial Attacks. Madry et al., ICLR 2017. \
> [R2] Obfuscated Gradients Give a False Sense of Security: Circumventing Defenses to Adversarial Examples. Athalye et al., ICLR 2019. \
> [R3] On Evaluating Adversarial Robustness. Carlini et al., 2019.
>
> **Q3**: "As the authors have trained models on MNIST and CIFAR-10, why not demonstrate Table 2 on CIFAR-10 (relatively small improvements as shown in Appendix)? According to Table 1 and Appendix E, the method performs better on MNIST, I suspect it will perform worse on large datasets(e.g., ImageNet)." \
> **A3**: To address the reviewer's concern,  we conducted **extra experiments on CIFAR-10** and obtained consistent results as follows. Similar to Table 2, with the prior knowledge of robustness of given models ($C > D > A > B$), we devised several heuristic baselines including: (a) $w_{c+d}$: ensemble PGD on models C and D only; (b) $w_{b+c+d}$: ensemble PGD on model B, C, D only; (c) $w_{a+c+d}$: ensemble PGD on models A, C and D only; (c) $w_{clip}$: clipped version of C\&W loss (threshold $\beta = 50$) to balance model weights in optimization as suggested in [R4]; (d) $w_{prior}$: larger weights on the more robust models, $w_{prior} = [w_A, w_B, w_C, w_D] = [0.2, 0.1, 0.4, 0.3]$; (e) $w_{static}$: the converged mean weights of min-max (APGDA) ensemble attack $w_{static} = [0.597, 0.042, 0.243, 0.118]$.
>
> |            | $Acc_A$ | $Acc_B$ | $Acc_C$ | $Acc_D$ | ASR$_{all}$ |
> |:----------:|:-------:|:-------:|:-------:|:-------:|:---------:|
> |   $w_{c+d}$  |  40.55  |  30.98  |   5.02  |   5.53  |   40.13   |
> |  $w_{b+c+d}$ |  42.12  |   1.63  |   5.93  |   4.42  |   51.63   |
> |  $w_{a+c+d}$ |  13.33  |  32.41  |   4.83  |   5.44  |   56.89   |
> |  $w_{clip}$  |  11.13  |   3.75  |   6.66  |   6.02  |   77.82   |
> |  $w_{prior}$ |  19.72  |   2.30  |   4.38  |   4.29  |   73.45   |
> | $w_{static}$ |   7.36  |   4.48  |   5.03  |   6.70  |   81.04   |
> |     avg    |  19.69  |   1.55  |   5.61  |   4.26  |   73.29   |
> |   minmax   |   7.21  |   2.68  |   4.74  |   4.59  |   84.36   |
>
> As we can see, our proposed method outperforms all heuristic baselines by a large margin on CIFAR-10 as well, which again verifies the power of dynamically optimizing domain weights during the attack generation process.
>
> As for the “relatively small improvement on CIFAR”, we kindly note that the *relative improvements* not only depend on the dataset but also depend on the model sets and attack strength ($\epsilon$) as shown in Figure 2 and Table A3/A4. For example, our min-max strategy would lead to a significant improvement if victim models have diverse attack difficulties with moderate attack strengths.
>
> To further alleviate reviewer’s concern, we conducted **new experiments on ImageNet** - 1000 eval images [R4]. Specifically, we considered four models - VGG16, ResNet34, EfficientNet-B0 [R5] and ViT-B/16 [R6] and used the cross-entropy loss function. We set the attack iteration as 7, $\epsilon=4/255$, step size as $1/255$  for attack generation. We set $\beta=50, \gamma=10$ for APGDA.
>
> |        | Acc_{VGG16} | Acc_{ResNet34} | Acc_{EfficientNet-B0} | Acc_{ViT-B/16} | ASR_all |  Lift |
> |:------:|:-----------:|:--------------:|:---------------------:|:--------------:|:-------:|:-----:|
> |  clean |    69.90    |      74.30     |         77.10         |      72.60     |    -    |   -   |
> |   avg  |    11.20    |      1.50      |          5.20         |      19.60     |  75.30  |   -   |
> | minmax |     3.50    |      2.10      |          3.10         |      5.30      |  92.40  | 22.7% |
>
> As we can see, APGDA achieves significant improvement for more advanced models on ImageNet. We hope these additional experiments address the reviewer's concerns/suspects on the effectiveness of our method.
>
> [R4] Universal Adversarial Training. Shafahi, et al., 2018. \
> [R5] Natural Adversarial Examples. Hendrycks et al., 2019. \
> [R6] EfficientNet: Rethinking Model Scaling for Convolutional Neural Networks. Tan et al., 2019. \
> [R7] An Image is Worth 16x16 Words: Transformers for Image Recognition at Scale. Dosovitskiy et al., 2020.

---

> > ### Author Response · Authors · 2021-08-25
> > **Looking forward to follow-up discussions**
> >
> > We thank the reviewer for taking precious time in checking our responses. We hope our answers and additional experiments address your concerns well. Specifically,
> >
> > - Q1/A1: Our work is very different from with Shafahi et al., 2018 - solving different problems with different methodologies.
> > - Q2/A2: We hesitate to call MNIST a poor dataset for studying adversarial examples according to previous works. Note that we also conducted experiments on CIFAR-10 with larger models. We also provided additional experiments on ImageNet.
> > - Q3/A3: We conducted additional CIFAR-10 and ImageNet experiments and the results are consistent.
> >
> > Any further follow-up discussions are highly appreciated!

---

> > ### Comment · Reviewer_6whq · 2021-08-29
> > **Reply to Authors**
> >
> > I thank the authors for their response, but I am still concerned about the novelty of this work.
> >
> > All in all, I am on the fence.

---

> > > ### Author Response · Authors · 2021-08-30
> > > **Response and look forward to clarification**
> > >
> > > Thank you very much for the follow-up comment “I thank the authors for their response, but I am still concerned about the novelty of this work. All in all, I am on the fence.”
> > >
> > > In the earlier phase, reviewer commented “It doesn't differ much from previous work (Universal Adversarial Training, Shafahi et al., 2018)”. However, we have clearly demonstrated at https://openreview.net/forum?id=xlNpxfGMTTu&noteId=MJjlArPtTL_ that our work is **significantly different from the referred work and is novel to the community**. **We sincerely wish that Reviewer 6whq can provide more specific comments on the novelty concern.** We are happy to answer any follow-up questions.
> > >
> > > From the authors’ perspective, our work is of sufficient novelty as recognized by Reviewer cLBV, YmFH, and xJB1 (who raised the score to 6).
> > >
> > > **At the attack side**, we provided a unified min-max attack framework for the first time with applications to ensemble attack, universal attack, and EoT attack. In our attack formulation, the use of self-learnable domain weights $\mathbf w$ yields practical benefits since it adjusts the model robustness or attack power automatically during training, which makes the learning process more effective and explainable (e.g., [Table 6, 7 and Figure A2]).
> > >
> > > **At the defense side**, the min-max attack idea is further leveraged to advance the defense setting (defense against multiple perturbation domains) and achieve the SOTA performance (Sec. 5). We hope the reviewer can take a complete and precise assessment on our contributions.
> > >
> > > **Lastly**, the proposed min-max design is significant as it is easily compatible with existing adversarial attack generation methods. For example, we can actually adapt our approach by introducing learnable domains to make the gradients optimized in a more efficient way (attacking the worst-case image thus leading to a better trade-off).  To demonstrate this significance, we have conducted new experiments following Reviewer xJB1’s suggestion. Specifically,  we follow the reviewer’s suggestion and pick “Double Targeted Universal Adversarial Perturbations” (DTA) [1] as a case study. We adapt our min-max formulation to the UAP  algorithm in [1] - the algorithm is provided in this anonymous link: ​​https://ibb.co/xDHqGBz (where the changes over DTA are highlighted in red). We conducted an experiment on CIFAR-10 with ResNet20 and followed the exact same settings as [1].
> > >
> > > | CIFAR-10 (ResNet20) | $S_0$ | $S_1$ | $S_2$ | $S_3$ | $S_4$ | $S_5$ | $S_6$ | $S_7$ | $S_8$ | $S_9$ | Avg. |
> > > |---------------------|:-----:|:-----:|:-----:|:-----:|:-----:|:-----:|:-----:|:-----:|:-----:|:-----:|:----:|
> > > | DTA                 |  76.0 |  83.2 |  85.5 |  85.9 |  78.0 |  78.6 |  83.7 |  68.8 |  76.2 |  86.6 | 80.3 |
> > > | DTA-PGDA (ours)     |  **82.6** |  **89.7** |  **91.0** |  **91.2** |  **81.1** |  **86.7** |  **89.2** |  **85.1** |  **90.2** |  **92.0** | **87.9** |
> > >
> > > As shown in the table, **our DTA-PGDA algorithm outperforms DTA [1] significantly across 10 targeted UAP scenarios ($S_0$ to $S_9$) used in [1] (please see [1]’s Table 1)**. This again verifies the effectiveness and high compatibility of our proposal.
> > >
> > > We hope our further clarifications are helpful. Any further discussions are welcome and appreciated!
> > >
> > > [1] Double targeted universal adversarial perturbations. Benz et al, 2020.

---

### Official Review · Reviewer_YmFH · 2021-07-17

**Rating:** 7
**Confidence:** 4

**Summary:**

This paper adapts the min-max problem proposed for multi-domain robust optimization [45] to the problem of generating adversarial attacks. They show how several adversarial attacks scenarios can be formulated using this min-max, for example they show that this min-max formulation can be used to find successful attacks against an ensemble of models. They also show how this min-max can also be used to train robust models against multiple $\ell_p$ attacks. They show experimentally that the approach outperforms several baseline in several setting.

**Limitations And Societal Impact:**

Yes

**Main Review:**

**Originality**:
This paper adapts and existing min-max problem that was proposed for multi-domain robust optimization [45] to the problem of generating adversarial attacks. This work clearly explains its contributions and put them into context of existing work. However, the author mention that some paper have explored min-max formulation in the context of adversarial attacks but don't cite them, for example one work related to problem eq. 5 that also uses a min-max formulation is (Bose et al. 2020).

**Quality**:
The claims of the paper are overall well supported. However It would be nice to have confidence intervals for the experimental results.
I really enjoyed the experimental section, I found Table 2 and Figure 1 quite interesting, showing the importance of having adaptive weights and how the weights acts like an attention mechanism to focus more on the hardest task. Table 4 is also quite informative.
The author address the time complexity and additional parameters in section 5.3.

**Clarity**:
The paper is well written and very clear.

**Significance**:
The idea of training against the worst-case scenario is very interesting, this could lead to much more universal attacker.

**Minor comments**:
- Typo L298: $\gamma=\infty$ for averaging case


**Additional references**:
- Bose et al. "Adversarial Example Games." (NeurIPS 2020).

**Time Spent Reviewing:**

3

---

> ### Author Response · Authors · 2021-08-10
> **Response to Reviewer YmFH**
>
> We thank the reviewer for the thoughtful review. We are glad that the reviewer recognizes our contributions and has positive comments. We will correct the minor typos and add discussion with the referred work (Bose et al., 2020) in Related Work.

---

### Official Review · Reviewer_xJB1 · 2021-07-18

**Rating:** 6
**Confidence:** 4

**Summary:**

This paper proposes a unified alternating projected gradient descent-ascent (APGDA) attack by considering the min-max operation for adversarial attack settings. The proposed method can be applied in generating model ensemble attacks, universal adversarial attacks, and robust attacks over data transformations. Extensive results show that the proposed method can achieve better attack performances for these three applications.

**Limitations And Societal Impact:**

No limitations and societal impact.

**Main Review:**

Strength:
- The proposed method is general in generating attacks when multiple domain losses are needed to be optimized together. The insights behind the method in smoothifying and balancing weights are very interesting for the adversarial attack with multiple domains.
- The paper is well written and easy to follow.
- Comprehensive experiments illustrate the strength of the proposed methods over general attack methods. The ablation study and the following analysis are insightful in improving the studied three tasks.

Weakness and comments:
- The method is not that novel as it simply applies non-convex min-max optimization for adversarial attacks. The proposed min-max optimization and concave regularizer are mostly proposed in existing works like [45].
- More importantly, a very similar idea for ensemble attack via universal perturbation is explored in [G] (e.g., equation 13 in [G]) even though it is more for black-box attack setting. Comparisons for ensemble model evasions and universal perturbations settings against [G] are very important and discussions about the differences in related work are at least needed.
- The proposed methods are not well compared with state-of-the-art attacks for the universal adversarial attacks. Generating universal adversarial perturbations (UAPs) has been extensively studied in both targeted [A,B,C] and non-targeted scenarios [D, E, F]. The authors should at least compare against one or two recent methods to demonstrate the effectiveness of the proposed methods.

[A] Double targeted universal adversarial perturbations, ACCV 2020

[B] Understanding adversarial examples from the mutual influence of images and perturbations, CVPR 2020

[D] Ask, acquire, and attack: Data-free UAP generation using class impressions, ECCV 2018

[E] Cross-domain transferability of adversarial perturbations, NeurIPS 2019

[F] Regional homogeneity: Towards learning transferable universal adversarial perturbations against defenses, ECCV 2020

[G] Min-Max Optimization without Gradients: Convergence and Applications to Adversarial ML, ICML 2020

**Time Spent Reviewing:**

3.5

---

> ### Author Response · Authors · 2021-08-10
> **Response to Reviewer xJB1**
>
> We thank the reviewer for thoughtful reviews and comments. We are glad that the reviewer believes that “smoothifying and balancing weights are very interesting for the adversarial attack with multiple domains”. We address the concerns as follows.
>
> **Q1**: The method is not that novel as it simply applies non-convex min-max optimization for adversarial attacks. The proposed min-max optimization and concave regularizer are mostly proposed in existing works like [45]. \
> **A1**: First, we admit that robust optimization over multiple domains has been studied in the literature like [45]. **However, our paper has a very different research focus compared with the optimization literature.** To our best knowledge, this is the first work to identify the power of min-max principle in  **attack design**  and achieve new state-of-the-art performance on multiple attack generation tasks. Like adversarial training (Madry et al, 2017), although our attack framework is derived from robust min-max optimization, its formulation and generalization to different attack types are new to the adversarial community, which are recognized as “novel” and “significant” by other reviewers (cLBV, YmFH).
>
> Meanwhile, we would like to respectfully mention that the seemingly 'direct' application of min-max optimization does not mean that our contributions are minor, instead it is highly non-trivial to demonstrate the generalizability and flexibility of our proposal in both attack design and robust training, and achieve superior performance in all examined scenarios. In both attack and defense, we show that the introduction of domain weights w is interpretable and has practical benefits since it adjusts the model robustness or attack power automatically during training, which makes the learning process more explainable. Finally, a significant experimental effort has been made to support our conceptual contributions.
>
> **Q2**: A very similar idea for ensemble attack via universal perturbation is explored in [G] (i.e., equation 13 in [G]) \
> **A2**: Thanks for pointing out the reference! We would like to highlight that Eq. (13) is not proposed in [G] but proposed in another arxiv paper (Wang et al., 2019b) [https://arxiv.org/pdf/1906.03563.pdf], cited prior to Eq. (13).
>
> We kindly bring the reviewer’s attention that this arXiv paper should not be compared with ours. And we are **very sure** that our min-max attack framework is new and proposed for the first time.
>
>
> **Q3**: The proposed methods are not well compared with state-of-the-art (SOTA) attacks for generating universal adversarial perturbations [A-F]. \
> **A3**: Thanks for providing the recent works on producing universal adversarial perturbations (UAP)! We will include them in the related work.
>
> First, UAP is merely one special application of our proposed min-max attack principle. Besides UAP, we explore the power of min-max principle in (1) ensemble attacks over multiple models, (2) robust attacks against different data transformations, as well as (3) adversarial training against multiple types of perturbation (Sec. 5.2).
>
> Second, we kindly remind the reviewer that the problem setups [A-F] (e.g., studying the transferability, using the proxy datasets) are different from ours, which is orthogonal to these works in producing universal adversarial perturbations. Specifically, we are tackling the problem from the min-max optimization perspective instead of specific algorithmic customization. For instance, in the algorithms proposed in [A, B], there is step $g_\delta \leftarrow \mathbb{E}_{x \sim B} [\Delta_\delta \mathcal{L}]$, which calculates the gradients in the average formulation. We can actually adapt our approach by introducing learnable domains to make the gradients optimized in a more efficient way (attacking the worst-case image thus leading to a better trade-off). We will include more discussions on how our proposal has compatibility with the other specific UAP algorithms.

---

> > ### Author Response · Authors · 2021-08-25
> > **Looking forward to follow-up discussions!**
> >
> > We thank the reviewer for spending precious time in checking our responses. We hope our responses address your concerns well. Specifically,
> >
> > Q1/A1: Our paper has a very different research focus compared with the optimization literature. To our best knowledge, this is the first work to identify the power of min-max principle in attack design and achieve new state-of-the-art performance on multiple attack generation tasks. \
> > Q2/A2: [G] borrowed the min-max formulation from another arxiv paper, which should not be considered. We are very sure that our min-max attack framework is new and proposed for the first time. It would be great if you could confirm with AC/SAC for our argument.\
> > Q3/A3: We clarify the differences between previous UAP works and this submission.
> >
> > Any further follow-up discussions are highly appreciated!

---

> > > ### Comment · Reviewer_xJB1 · 2021-08-28
> > > **Thanks the response, but my concerns remain**
> > >
> > > Thank the authors for the detailed response and active discussions. However, my concerns still remain especially for the second and third questions.
> > > - Some of the contributions have already been taken by [G] which was published in ICML 2020, even though they used the idea proposed in the arxiv paper [H: https://arxiv.org/pdf/1906.03563.pdf]. Also, the arxiv paper [H] is released in 2019. You cannot claim that your attack is new and you are the first to propose using a very similar idea. This is not fair for [G] and [H].
> > > - I do not see a straightforward way to directly combine the proposed method with existing [A-F] without affecting the attack performance. To make your claim convincing, please provide detailed algorithms and experimental results showing "your method + [A, B] > some of [A-F]". Alternatively, results showing "your method > some of [A-F]" can also be very helpful.
> > >
> > > Overall I will keep my original score for now.

---

> > > > ### Author Response · Authors · 2021-08-29
> > > > **Further clarifications with additional experiments**
> > > >
> > > > Thanks for the follow-up discussions! We would like to provide more clarifications on your concerns.
> > > >
> > > > **"Some of the contributions have been taken by [G]. It is not fair to state that you are the first to propose given the existence of [G] and [H]"**
> > > >
> > > > **Comparison with [G]: No overlapping**.
> > > >
> > > > Reference [G] focused on the development of **zeroth-order** min-max optimization method and theory. To demonstrate the effectiveness of the proposed method, a min-max universal attack formulation was borrowed from [H] with a clear citation in the equation Eq. (13) of [G]. To the best of the authors' knowledge, except the equation Eq. (13) of [G] cited by [H] (**not a contribution of [G]**), we did not see any overlapping place/contribution with [G].
> > > >
> > > > We sincerely hope that the reviewer can provide **specific and precise evidence** to justify the comment 'Some of the contributions have already been taken by [G]'. This way, we could better understand your concern, and we are certainly happy to address it if that can be specified.
> > > >
> > > > **Comparison with [H]:** \
> > > > We highly recommend the reviewer to check with AC and SAC about the correctness of our statement that reference [H] should not be taken into account when judging the significance of our contributions. Due to the anonymity rules, we cannot provide more information. But we are pretty sure that this submission is the first work to propose this idea (**NOT developing a similar idea with previous works**). ​​We thank the reviewer in advance for this extra communication overhead. Really appreciate it!
> > > >
> > > > **"Comparison of existing UAP works"**
> > > >
> > > > We would like to thank the reviewer for providing recent UAP works and suggesting comparisons with our work. Our response will be unfolded into two aspects.
> > > >
> > > > **First**, to justify our previous statement  ‘our work is orthogonal to these works in producing universal adversarial perturbations’, we follow the reviewer’s suggestion and pick the “Double Targeted Universal Adversarial Perturbations” (DTA) [A] as a case study. We  adapt our min-max formulation to the UAP  algorithm in [A]. We call our algorithm DTA-PGDA since the projected gradient ascent is used to optimize the min-max objective. Specifically, we introduce the domain weight $\mathbf w$ in the objective. Before gradient calculating, we first conduct a projected gradient ascent step to update the $\mathbf w$ for us to put more weights on more difficult images (inner maximization). Then, the projected gradient descent is conducted on weighted loss to obtain the perturbation updates (outer minimization). For better understanding, we provide our DTA-PGDA algorithm in this anonymous link: ​​https://ibb.co/xDHqGBz (where the changes over DTA are highlighted in red).
> > > >
> > > > We adopted the public implementation (https://github.com/phibenz/double-targeted-uap.pytorch) provided by [A] and implemented our algorithm (DTA-PGDA). We conducted an experiment on CIFAR-10 with ResNet20 and followed the exact same settings as [A].
> > > >
> > > > | CIFAR-10 (ResNet20) | $S_0$ | $S_1$ | $S_2$ | $S_3$ | $S_4$ | $S_5$ | $S_6$ | $S_7$ | $S_8$ | $S_9$ | Avg. |
> > > > |---------------------|:-----:|:-----:|:-----:|:-----:|:-----:|:-----:|:-----:|:-----:|:-----:|:-----:|:----:|
> > > > | DTA                 |  76.0 |  83.2 |  85.5 |  85.9 |  78.0 |  78.6 |  83.7 |  68.8 |  76.2 |  86.6 | 80.3 |
> > > > | DTA-PGDA (ours)     |  **82.6** |  **89.7** |  **91.0** |  **91.2** |  **81.1** |  **86.7** |  **89.2** |  **85.1** |  **90.2** |  **92.0** | **87.9** |
> > > >
> > > > As shown in the table, **our DTA-PGDA algorithm outperforms DTA [A] significantly across 10 targeted UAP scenarios ($S_0$ to $S_9$) used in [A] (please see [A]’s Table 1)**. This again verifies the effectiveness of our proposal.
> > > >
> > > > **Second**, we also would like to highlight that UAP is merely a special application of using our proposed min-max attack framework. We sincerely wish reviewers not to ignore our other contributions in the design of ensemble attack, expectation over transformation attack, and defense against multiple perturbation domains with a clear SOTA performance.
> > > >
> > > > We hope our further clarifications and additional experiments are helpful. Any further discussions are welcome and appreciated!

---

> > > > > ### Comment · Reviewer_xJB1 · 2021-08-29
> > > > > **Concerns addressed**
> > > > >
> > > > > Thanks for the further clarification and additional results. I have raised my score.

---

> > > > > > ### Author Response · Authors · 2021-08-30
> > > > > > **Thank you!**
> > > > > >
> > > > > > Thank you very much for raising the score. We are very glad to learn that our responses have made the points clearer. We will revise the paper according to your insightful comments.

---

### Official Review · Reviewer_CtyL · 2021-07-19

**Rating:** 4
**Confidence:** 4

**Summary:**

The paper provides a general min-max framework to formulate several popular problems in the literature including Ensemble attack over multiple models, Universal Perturbation over multiple examples, and adversarial attack over data transformations. For each case, a standard first-order approach converging to a stationary point of the problem is provided. The effectiveness of the min-max approach compared to the standard empirical risk minimization has been demonstrated by extensive experiments.

**Limitations And Societal Impact:**

The authors provide ample justification for the potential benefits of their work in safety-critical AI applications.

**Main Review:**

Major Concerns:
- While the effectiveness of the proposed min-max framework for solving several problems over the regular empirical risk minimization is shown through the experiments, the novelty of such formulations is not clear for the reviewer. There are many papers in the literature using the same min-max formulation for solving such problems, especially for attack generation (see [1]).

- Moreover, almost all formulations, lemmas, and theorems in the paper use the basic idea that the max of finite terms can be written as the maximization of a convex combination of the terms over a simplex. Mathematically speaking:
Max (x_1, x_2, …, x_n) = max_{w} w_1 x_1 + w_2 x_2 + … w_n x_n        s.t \sum_{i=1}^n w_n x_n = 1, x_i >= 0 for all I.

It seems the common element among different investigated problems is the min-max formulation (which is not novel) and the mentioned trick. Thus, it cannot be considered a framework in the reviewer's opinion.

- From an optimization perspective, the contribution of the paper is limited. The provided algorithms are basically different variations of the projected gradient descent ascent algorithm.

Strengths:
- The experiments show that the robust formulation (min-max) of the attack generation problem works far better than the average case.

**Time Spent Reviewing:**

8

---

> ### Author Response · Authors · 2021-08-10
> **Response to Reviewer CtyL**
>
> We thank the reviewer for thoughtful reviews and comments. We address the concerns as follows.
>
> **Q1**: Limited novelty from an optimization perspective. There are many papers in the literature using the same formation to solve such problems (see [1]) \
> **A1**: We **respectfully disagree** with the limited novelty of our work.
>
> First, if [1] refers to “Square attack: A query-efficient black-box adversarial attack via random search, ECCV’20” in our reference section, then it is significantly different from ours. If the reviewer meant [1] for another paper, please provide it.
>
> Second, we admit that min-max optimization is a well-studied topic in the optimization literature. However, this does not restrict our novelty in the field of adversarial machine learning.
> This is the **first solid step** to explore the power of min-max principle in the **attack design** and achieve superior performance on multiple attack tasks: (1) ensemble attacks over multiple models, (2) universal perturbations against multiple images, and (3) robust attack against different data transformations. We also showed that the application of min-max attack can advance (4) adversarial training against multiple types of perturbation, and achieves a new performance benchmark compared to the state-of-the-art MSD method (Sec. 5.2). The unified min-max attack is new to the adversarial community, which is recognized as “novel” and “significant” by other reviewers (cLBV, YmFH).
>
> Third, we would like to respectfully mention that the seemingly "direct" application of min-max optimization does not mean that our contributions are minor, instead it is highly non-trivial to demonstrate the generalizability and flexibility of our proposal in both attack design and robust training, and achieve superior performance in all examined scenarios. In both attack and defense, we show that the introduction of domain weights w is interpretable and has practical benefits since it adjusts the model robustness or attack power automatically during training, which makes the learning process more explainable. Finally, a significant experimental effort has been made to support our conceptual contributions.
>
> **Q2**: All formulations, lemmas, theorems use a basic idea/trick thus this work cannot be treated as a framework. \
> **A2**: Thanks for your comment. The probabilistic simplex (together with its squared $\ell_2$ regularization) provides us a novel view to attack multiple models, images, data transformations in a unified manner through the learnable importance weights. We strongly believe that this is new to the community of adversarial machine learning, which is recognized by other reviewers (cLBV, YmFH). Furthermore, the min-max attack idea was further leveraged to advance the defense setting (defense against multiple perturbation domains) and achieve the new state-of-the-art (SOTA) performance. This again justifies that our proposal can be treated as a **novel framework** to explore and exploit adversarial robustness.

---

> > ### Author Response · Authors · 2021-08-25
> > **Looking forward to follow-up discussions!**
> >
> > We thank the reviewer for spending precious time in checking our responses. We hope our responses address your concerns on the novelty of this work. We are pleased to take any future questions!
> >
> > - We respectfully disagree that our formulation is already studied before. **If [1] refers to “Square attack: A query-efficient black-box adversarial attack via random search, ECCV’20” in our reference section, then it is significantly different from ours. If the reviewer meant [1] for another paper, please provide the reference.**
> > - We admit that min-max optimization is a well-studied topic in the optimization literature. However, this does not restrict our novelty in the field of adversarial machine learning. This is the first solid step to explore the power of min-max principle in the attack design and achieve superior performance on multiple attack tasks. **Like the seminar work in this domain - adversarial training & PGD (Madry et al., 2017), although there is nothing new from the optimization perspective (min-max optimization or gradient descent), the ways of using this optimization tool to generate perturbations and robustify models yields significant empirical improvement and are highly impactful.**
> >
> > Any follow-up discussions are highly appreciated.

---

> > ### Comment · Reviewer_CtyL · 2021-08-27
> > **Not Enough Contribution**
> >
> > - Regarding the novelty, I do not think this is the first solid work using minimax formulation to generate adversarial attacks. For instance, see [1], where a joint detector/classifier is designed to handle adaptive attacks to solve a minimax problem.  Such minimax formulations have been widely used in practice since especially after [2] published in 2017.
> > - Sections 2 and 3 describe the existing formulations for modeling adversarial attacks (the authors mentioned [3], which includes very similar formulations). However, it is not clear what are the benefits of algorithms proposed in this paper compared to the ones described in [3].
> > - Writing a finite max problem as an optimization problem over a probability simplex is a fairly simple technique and cannot be considered a sufficient contribution for a NeurIPS paper.
> >
> > [1] Sheikholeslami, Fatemeh, Ali Lotfi, and J. Zico Kolter. "Provably robust classification of adversarial examples with detection." International Conference on Learning Representations. 2020
> > [2] Madry, Aleksander, et al. "Towards deep learning models resistant to adversarial attacks." arXiv preprint arXiv:1706.06083 (2017).
> > [3] Q. Qian, S. Zhu, J. Tang, R. Jin, B. Sun, and H. Li. Robust optimization over multiple domains. 454 CoRR, abs/1805.07588, 2018.

---

> > > ### Author Response · Authors · 2021-08-27
> > > **Enough contribution: The mentioned works have different scopes than ours**
> > >
> > > Thank you for your response. However, we respectfully disagree with "not enough contribution".
> > >
> > > **Comparison with [1]:** \
> > >  Reference [1] and the adaptive attack were used as evidence against us on the lack of novelty. However, we kindly bring reviewer’s attention that this is **significantly** different from our work.
> > >
> > > The main theme of [1] is to develop a method of jointly training a provably robust classifier and detector. Yes, a min-max based adaptive attacker’s objective (Eq. (12) of [1]) was introduced to craft perturbation so that it simultaneously evades detection and causes misclassification. However, this min-max formulation is different from ours. In Eq. (12) of [1], only the max-principle was used for the design of adaptive attacks. This is just a special case of ours (Line 84 - 87). In our attack formulation (e.g., Eq. (5)-(7))), self-learnable domain weights $\mathbf w$ were used. This is critical as we showed in [Table 6, 7 and Figure A2] that the introduction of $\mathbf w$ is interpretable and has practical benefits since it adjusts the model robustness or attack power automatically during training, which makes the learning process more effective and explainable. **Moreover**, our proposed mix-max formulation has been applied to universal attack, ensemble attack and EoT attack and achieved superior performance.  **Furthermore**, we leverage the min-max attack idea to advance the defense setting (defense against multiple perturbation domains) and achieve the SOTA performance (Sec. 5).
> > >
> > > Based on the above evidence and to the best of our knowledge, our min-max attack design has not been widely used in practice. We are also happy to answer any follow-up question, if the reviewer can provide more evidence as to why the proposed min-max attack has widely been used in practice prior to our work.
> > >
> > > **Comparison with [3]:** \
> > > The main contribution of [3] lies in the optimization theory part. Please note that ‘robust optimization over multiple domains’ is in fact quite natural to the optimization community. As we stated in Line 77-81, the incorporation of strongly concave regularization at the inner-level maximization was suggested and used in the min-max optimization community. We have clearly stated in Line 82 that we indeed followed the formulation of robust optimization over multiple domains.
> > >
> > > Compared to [3], our main contribution lies at the application to the adversarial attack and defense domain. **It is our contribution** to adapt, apply, and extend such formulation for the unified design of adversarial attack and defense (against mixed attacks). As we have clarified in your first question, the benefits of our algorithm are clear.
> > >
> > > **`Simplicity’ does not imply the lack of novelty:** \
> > > The optimization technique is fairly simple, however, the simplicity brings in a unified attack generation framework and significant empirical improvement over baseline methods in attack and defense. As we clarified in the previous questions, we achieved the new state-of-the-art (SOTA) performance on ensemble attack, universal attack, and expectation over transformation attack. And the min-max attack idea is further leveraged to advance the defense setting (defense against multiple perturbation domains) and achieve the SOTA performance (Sec. 5). We hope the reviewer can take a complete and precise assessment on our contributions.

---

### Official Review · Reviewer_cLBV · 2021-07-22

**Rating:** 6
**Confidence:** 4

**Summary:**

The paper studies a min-max formulation for adversarial attacks over multiple domains, which covers attacking ensembling attacks, multiple threat models, and universal perturbations. They propose a regularized formulation and show improvements performance on these tasks.

**Limitations And Societal Impact:**

I did not find a discussion of the limitations of their work. It would be nice if the authors described some possible properties of their approach that may not be a strict improvement over the compared baselines.

**Main Review:**

The proposed regularized formulation of adversarial attack appears to improveme the attack performance in all three settings. Empirically, the paper seems sufficient. However, the paper suffers greatly with issues in clarity. The paper as a whole could be improved greatly, and two parts that stand out are:

1. The paper introduces their formulation as a problem of robust learning. This is somewhat misleading, as the "robust learning" in this paper is referring to a robust formulation of the adversarial attack, and not for updating the model parameters (i.e. "learning"). The latter is what I see most commonly in the literature for robust learning, and it is not clear to me that this is a formulation for adversarial attacks until Section 3.

2. The results are not clear and require a lot of parsing. For example, the text refers to ensemble baselines, but it is unclear what in the table this refers to (I assumed it was the ones with "avg" under "opt", since the table captions refer to both attacks as ensemble attacks). There is a column in the table that is labeled "Lift" which seems to suggest an amount of improvement, however it is not clear to me what number or metric this is supposed to be improving and I could not find any explanatory text in either the caption or the text.

Overall, the proposed formulation and the corresponding results seem alright. I do like how the authors studied a simple, generalized regularized formulation and evaluated it in multiple attack settings. However this was only after struggling to parse and understand the paper, the presentation of which was not very clear to me.

Other minor questions/comments:

The convergence analysis (Theorem 1) appears to make very strong assumptions on the problem. Specifically, it is assumed that the problem has L-Lipschitz continuous gradients. Is this really a reasonable assumption for the setting considered in this paper (i.e. classification in deep networks)?

There are some recent works that improved upon the multi-perturbation robustness setting (over MSD) that would be nice to compare to, i.e. "Learning to generate noise for multi-attack robustness" (this is recent work, and so I did not hold this against the authors, but it would be nice to include in the final version).

**Time Spent Reviewing:**

4

---

> ### Author Response · Authors · 2021-08-10
> **Response to Reviewer cLBV**
>
> Thanks for your thoughtful reviews and comments. We are glad to see the reviewer like our simple formulation and sufficient studies. We also appreciate the suggestions on improving the clarity of the presentation.
>
> **Q1**: “Robust learning” in this paper is referring to a robust formulation of the adversarial attack, and not for updating the model parameters (i.e., “learning”). \
> **A1**: Thanks for pointing this out! We agree that the term “robust learning” might cause some confusion as it does not refer to the standard robust training or adversarial learning. We will replace the term “robust learning” with “robust optimization” in the revision.
>
> **Q2**: Ensemble PGD attack in the text but “avg” under “opt” in the table. The “Lift” metric is not introduced in the text. \
> **A2**: Thanks for the suggestion. (1) We adopt ensemble PGD following the work [R1] which refers to the average formulation in the paper. We agree that it might cause confusion as both average or min-max PGD are ensemble attacks. Therefore, we would like to replace “ensemble PGD” with “average PGD” to make it clearer. (2) For the “Lift” metric, which refers to the relative improvement, the concrete definition is:
> $Lift = (metric_A - metric_B) / metric_B * 100\\% $ \
> We will add explanatory texts for this in the caption or the main paragraphs to make it clear.
>
> [R1] Iterative ensemble adversarial attack. Liu et al., 2018.
>
> Thanks again for the suggestions to improve the clarity. We apologize for not stating some points clearly. We will clarify those points that might cause confusion or be hard to parse in the revision.
>
> \
> Response to minor questions/comments:
>
> **Q3**: Strong assumptions in Theorem 1 ($L$-Lipschitz continuous for function $F$) \
> **A3**: Thanks for the comment. First, we believe that the $L$-smoothness is a common assumption used in attack generation; see [Assumption 1; R2] and [R3]. This is because although the original neural network is non-convex, however, after restricting to a small neighborhood ($\ell_p$-norm constraint), then the variation of the Hessian is bounded, enabling the $L$-smoothness within a local neighborhood of an input. Second, from the perspective of theoretical analysis, the L-smoothness is used as the standard assumption to proceed with the convergence rate analysis for the **nonconvex** min-max problem, as shown in [32] and [34] that we pointed out in Line 163.
>
> [R2] On the Convergence and Robustness of Adversarial Training. Wang et al., ICML 2019. \
> [R3] Robustness via curvature regularization, and vice versa. Moosavi-Dezfooli et al., CVPR 2019.
>
> **Q4**: Discussion with recent work “Learning to generate noise for multi-attack robustness” \
> **A4**: Thanks for providing the reference! We will include the discussions and comparisons in the final version (defense section).

---

### Official Review · Reviewer_ynwW · 2021-07-24

**Rating:** 5
**Confidence:** 5

**Summary:**

This paper proposed a APGDA attack method, in which the model ensemble attack, universal attack, and robust attack are unified. The authors proved their APGDA's superiority in theory, ${i.e.}$, convergence rate and strong attacking ability. All of the mentioned tasks (ensemble attack, universal attack, robust attack) are investigated through experiments.

**Limitations And Societal Impact:**

The authors have addressed the limitations.

**Main Review:**

$\textbf{Cons:}$

(1) According to Table 1,  it is clear that the proposed APGDA outperforms the compared methods in ${ASR}_{all}$ metric (for ensemble attack task).
However, I found that in the $ACC_B$ metric, the proposed method performs worse than the comparisons under each setting, indicating the possibility of error due to the model selection and causing my great concern. The same problem appears in Table A3 and Table A4.

(2)In addition, I think the experiments are not conducted fully. For example, the author conducted experiments (ensemble attacks) on MNIST with model ABCD but forgot to test EFH as tested on CIFAR-10. In Table A4's caption, all of the learning rate and regularization factors are the same as those in Table A3, but why did you change the parameter $\epsilon$ (0.05, 0.03 in $l_{\inf}$, and 2.0, 1.5 in $l_2$)?

(3)The proof of Theorem 1 seems not finished?

(4) When it comes to universal attack, I am confused that the PGD attack is not designed for "universal attack". Thus, comparing with PGD can not demonstrate the performance of the proposed APGDA.

(5)I am scared that the same concern as I mentioned in issue(1) occurs again according to Table 5.

(6) The noise size of the attacking methods has not been fully considered. I think the corresponding results will make this work more solid.

**Time Spent Reviewing:**

about 4 hours

---

> ### Author Response · Authors · 2021-08-10
> **Response to Reviewer ynwW**
>
> We thank the reviewer for the thoughtful comments. We would like to clarify some misunderstandings and address the concerns as follows.
>
> **Q1:** For Table 1, the proposed method performs worse in the $ACC_B$ metric. The same problem appears in Table A3, A4, and Table 5. \
> **A1:** Instead of treating worse $ACC_B$ metric as a weakness or “potential error due to model selection”, we believe, quite on the contrary, it clearly demonstrates the merit of the proposed APGDA algorithm - attacking against the **worst-case** environment that an adversary encounters while striking a balance with the easily-attacked environments. More specifically, in Table 1 ($\ell_\infty$ attack), since models C, D are more difficult to attack, APGDA wisely handles the worst case {C, D} by slightly sacrificing the performance on {A, B}: 31.47% ASR improvement on {C, D} vs. 0.86% degradation on {A, B}. We would like to refer the reviewer to Figure 1 for better explanations for this insightful phenomenon, as recognized by Reviewer YmFH.
>
> The above empirical results show that compared to the averaging strategy, our min-max approach assigns attack strengths to multiple domains dynamically based on the attack difficulty and achieves a much better trade-off. There is no free lunch to achieve significant attack improvements across all different domains since the attacker shares the same attack setting (attack iteration $T$, perturbation magnitude $\epsilon$, and attack step size $\alpha$) in these different domains. Note that in the paper, we have provided detailed explanations why our proposed min-max approach leads to a better trade-off (Line 120-128, Line 185-210).
>
> **Q2.1**: The author conducted experiments (ensemble attacks) on MNIST with model ABCD but forgot to test EFH as tested on CIFAR-10. \
> **A2.1**: We didn’t conduct experiments on more complicated models (e.g. ResNet50, GoogLeNet) on MNIST due to the simplicity of this dataset. We believe that very large models are unnecessary for evaluation on MNIST since (1) A-D models can already achieve very competitive performance (99.5%), and (2) more complicated models easily lead to overfitting issues on MNIST. To further alleviate the reviewer's concern we conduct a new experiment to evaluate the model set {A, B, E, F} on MNIST: $\ell_\infty$ attack ($\epsilon$ = 0.2).
>
> |        | $Acc_A$ | $Acc_B$ | $Acc_E$ | $Acc_F$ | ASR$_{\mathrm{all}}$ |
> |:------:|:-------:|:-------:|:-------:|:-------:|:-----------:|
> |   avg  |   2.38  |   0.15  |  38.51  |  46.03  |    45.32    |
> | minmax |   3.54  |   0.44  |   6.38  |   9.17  |    86.28    |
>
> As we can see, even in the presence of larger models E and F, our proposed min-max attack still outperforms the baseline with a large margin, consistent with our results in the submission.
>
> **Q2.2**: In Table A4's caption, all of the learning rate and regularization factors are the same as those in Table A3, but why change the parameter $\epsilon$? \
> **A2.2**: We change the parameter $\epsilon$ since different models are associated with different attacking difficulties. For larger models E, F, H, it is easier to generate adversarial perturbations, and thus we decrease the maximum perturbation magnitude from 2.0 to 1.5 ($\ell_2$ attack) and from 0.05 to 0.03 ($\ell_\infty$ attack). And changing $\epsilon$ enables us to characterize a complete picture of the $\ell_p$ attack diversity; Please refer to Figure 2 for a complete performance curve with varied epsilons.
>
> **Q3**: The proof of Theorem 1 seems not finished? \
> **A3**: The proof of Theorem 1 is finished. Note that problem (4) is in a special form of min-max optimization, where the inner problem with respect to (w.r.t.) $w$ is a strongly-concave maximization problem, with $\gamma$-Lipschitz continuous gradients. The only assumption that we impose on Theorem 1 is its $L$-smoothness w.r.t. the outer variable $\boldsymbol \delta$. However, the $L$-smoothness is a common assumption in attack generation; see [Assumption 1; R1] and [R2]. Now, given the smooth non-convex outer minimization problem plus the strongly concave inner maximization problem, the convergence of alternating gradient descent-ascent method can then be derived following the standard optimization literature, like [32] or [34] that we pointed out in Line 163. The proof is thus complete. We will add more proof details in the appendix for ease of understanding in the revision.
>
> [R1] On the Convergence and Robustness of Adversarial Training. Wang et al., ICML 2019. \
> [R2] Robustness via curvature regularization, and vice versa. Moosavi-Dezfooli et al., CVPR 2019.
>
> **Q4**: PGD attack is not designed for “universal attack” so the performance cannot show the performance of proposed APGDA. \
> **A4**: We would like to clarify that the “PGD” that we refer to is not the standard single-image PGD attack [R3], instead it follows the **general PGD optimization principle** to minimize the average of attack losses across multiple images for universal perturbation generation (see Eqn. (2) in [R4]).
> In the revision, we will make the above point clearer and encourage the reader to treat PGD (projected gradient descent) as an optimization method instead of a specific attack method. As shown in [R4], PGD over the averaging multi-image attack loss leads to significantly better attack success rate and efficiency than other approaches (e.g, DeepFool in [R5], Adam, MSDG, SGD optimization on the average formulation [R4]). In our work, we demonstrate the effectiveness of min-max PGD (or APGDA) in building universal perturbations.
>
> [R3] Towards Deep Learning Models Resistant to Adversarial Attacks. Madry, et al., 2018. \
> [R4] Universal Adversarial Training. Shafahi, et al., 2018. \
> [R5] Universal Adversarial Perturbations. Moosavi-Dezfooli, et al., 2017.
>
> **Q5**: The noise size of the attacking models has not been fully considered. \
> **A5**: We actually **have included** the results in the paper: Figure 2 and A1. As the perturbation radius $\epsilon$ varies, we also observe that the ASR of min-max strategy is consistently better or on par with the average strategy (Line 209-210).

---

> > ### Author Response · Authors · 2021-08-25
> > **Looking forward to follow-up discussions!**
> >
> > We thank the reviewer for taking precious time in checking our responses. We hope our answers and additional experiments address your concerns well. Specifically,
> > - Q1/A1: We explained why it is infeasible to achieve better performance for all domains. This observation should not be treated as concerns or weakness, quite on the contrary, it clearly demonstrates the merit of the proposed APGDA algorithm.
> > - Q2/A2: We conducted new experiments with larger model on MNIST and the performance is consistent. We changed the parameter $\epsilon$ since different models are associated with different attacking difficulties.
> > - Q3/A3: The proof of Theorem 1 is complete.
> > - Q4/A4: We explained why PGD is a good baseline for universal attack and provided some references.
> > - Q5/A5: We actually had included the results in the paper (Figure 2 and A1).
> >
> > Any further follow-up discussions are highly appreciated!

---

### Author Response · Authors · 2021-08-10
**General Response**

We would like to thank the reviewers again for their thoughtful reviews and valuable comments. In what follows, we summarize the major concerns from the reviewers and our responses.

**Clarification on novelty** [Reviewers CtyL, xJB1] \
Our paper has a very different research focus compared with the robust optimization literature. To our best knowledge, this is the **first solid step to explore the power of min-max principle in the attack design and achieve superior performance on multiple attack tasks**: (1) ensemble attacks over multiple models, (2) universal perturbations against multiple images, and (3) robust attack against different data transformations. We also showed that the application of min-max attack can advance (4) adversarial training against multiple types of perturbation, and achieves a new performance benchmark compared to the state-of-the-art MSD method (Sec. 5.2). The unified min-max attack is new to the adversarial community. Like adversarial training (Madry et al, 2017), although our proposal is derived using robust optimization techniques, its formulation and generalization to different attack types are new to the adversarial community, which are recognized as “novel” and “significant” by other reviewers (cLBV, YmFH).

Meanwhile, we would like to respectfully mention that the seemingly 'direct/simple' application of min-max optimization does **not** mean minor contribution, instead, it is highly non-trivial to demonstrate the generalizability and flexibility of our proposal in both attack design and robust training, and achieve superior performance in all examined scenarios. In both attack and defense, we show that the introduction of domain weights w is interpretable and has practical benefits since it adjusts the model robustness or attack power automatically during training, which makes the learning process more explainable. Finally, please note that we have made a significant experimental effort to support our conceptual contributions.

**New experiments suggested by Reviewers ynwW, 6whq**
- [To Reviewer ynwW] We have added the experiment on MNIST with larger models (models E, F: VGG16, and ResNet50), showing that our proposed min-max attack outperforms the baseline with a large margin, which is consistent with our results in the submission.
- [To Reviewer 6whq] We have added comparisons with the heuristic baselines on CIFAR-10 and ensemble attacks (avg vs min-max) comparisons on ImageNet with four models: VGG16, ResNet34, EfficientNet-B0, and ViT-B/16. Additional experiments verify that our proposed method works very well for larger datasets which should address the reviewer's suspects on the effectiveness of APGDA.

---

### Decision · Program_Chairs · 2021-09-27

**Decision:**

Accept (Poster)

**Comment:**

This review results for this paper are borderline. While the reviewers appreciate some of the positive aspects of the paper (such as the formulation and the comprehensive numerical experiments), they have reservations about the novelty and the presentation of parts of the paper. However, regarding the contribution of the paper, the authors' argument about the novelty of 1909.13806 (published) w.r.t. 1906.03563 (unpublished) is reasonable and may justify the contribution.